# Cross-species genomic landscape comparison of human mucosal melanoma with canine oral and equine melanoma

Kim Wong [1], Louise van der Weyden[1], Courtney R. Schott[2], Alastair Foote[3], Fernando Constantino-Casas[4], Sionagh Smith[5], Jane M. Dobson[4], Elizabeth P. Murchison [4], Hong Wu[6], Iwei Yeh[6], Douglas R. Fullen[7], Nancy Joseph[6], Boris C. Bastian[6], Rajiv M. Patel[7], Inigo Martincorena[1], Carla Daniela Robles-Espinoza[1,8], Vivek Iyer[1], Marieke L. Kuijjer[9,10,11], Mark J. Arends[12], Thomas Brenn[1,13], Paul W. Harms[7], Geoffrey A. Wood [2] & David J. Adams[1]

Mucosal melanoma is a rare and poorly characterized subtype of human melanoma. Here we perform a cross-species analysis by sequencing tumor-germline pairs from 46 primary human muscosal, 65 primary canine oral and 28 primary equine melanoma cases from mucosal sites. Analysis of these data reveals recurrently mutated driver genes shared between species such as *NRAS*, *FAT4*, *PTPRJ*, *TP53* and *PTEN*, and pathogenic germline alleles of *BRCA1*, *BRCA2* and *TP53*. We identify a UV mutation signature in a small number of samples, including human cases from the lip and nasal mucosa. A cross-species comparative analysis of recurrent copy number alterations identifies several candidate drivers including *MDM2*, *B2M*, *KNSTRN* and *BUB1B*. Comparison of somatic mutations in recurrences and metastases to those in the primary tumor suggests pervasive intra-tumor heterogeneity. Collectively, these studies suggest a convergence of some genetic changes in mucosal melanomas between species but also distinctly different paths to tumorigenesis.

[1] Wellcome Sanger Institute, Wellcome Trust Genome Campus, Hinxton, Cambridge CB10 1SA, UK. [2] Department of Pathobiology, University of Guelph, 50 Stone Road E., Guelph, ON N1G 2W1, Canada. [3] Rossdales Equine Hospital and Diagnostic Centre, High Street, Newmarket, Suffolk CB8 8JS, UK. [4] Department of Veterinary Medicine, Cambridge Veterinary School, University of Cambridge, Cambridge CB3 0ES, UK. [5] The Royal (Dick) School of Veterinary Studies and The Roslin Institute, Easter Bush Campus, Midlothian EH25 9RG, UK. [6] Departments of Dermatology and Pathology, University of California, San Francisco, CA 94143, USA. [7] Departments of Pathology and Dermatology, University of Michigan Medical School, 3261 Medical Science I, 1301 Catherine, Ann Arbor, MI 48109-5602, USA. [8] Laboratorio Internacional de Investigación sobre el Genoma Humano, Universidad Nacional Autónoma de México, Campus Juriquilla, Blvd Juriquilla 3001, Santiago de Querétaro 76230, Mexico. [9] Department of Biostatistics, Harvard T.H. Chan School of Public Health, Boston, MA 02215, USA. [10] Department of Biostatistics and Computational Biology, Dana-Farber Cancer Institute, Boston, MA 02215, USA. [11] Centre for Molecular Medicine Norway (NCMM), Nordic EMBL Partnership, Faculty of Medicine, University of Oslo, 0349 Oslo, Norway. [12] University of Edinburgh, Division of Pathology, Centre for Comparative Pathology, Cancer Research UK Edinburgh Centre, Institute of Genetics & Molecular Medicine, Western General Hospital, Crewe Road South, Edinburgh EH4 2XR, UK. [13] Department of Pathology and Laboratory Medicine, Cumming School of Medicine and Arnie Charbonneau Cancer Institute, University of Calgary, Calgary T2L 2K8, Canada. Correspondence and requests for materials should be addressed to D.J.A. (email: da1@sanger.ac.uk)

In European-descent populations, the majority of melanomas develop as a result of UV light exposure on hair-bearing skin, most commonly the trunk in men and the legs in women. Other subtypes of melanoma include uveal melanoma, which develops in the eye and acral lentiginous melanoma, which forms on locations such as the palms of the hands and soles of the feet[1]. In a small proportion of cases, melanoma may also develop at mucosal sites such as in the lining of the anal canal, vulva, vagina, urethra and in the oral and nasal cavities[2]. This type of melanoma represents around 1–2% of all cases, but comprises a higher proportion of diagnoses in non-white populations, where cutaneous melanoma is less common[2]. Intriguingly, while cutaneous melanoma is more common in men, mucosal melanoma is as much as two times more prevalent in women, due to the occurrence of vaginal/vulvar melanomas[3], particularly in the sixth decade and beyond[4]. While hundreds of genomes of cutaneous melanoma have now been sequenced, defining a landscape that is replete with UV-induced mutations and driver mutations in genes such as BRAF, the genomes of tumors that develop at mucosal sites have not been as well characterized[5–8]. To date, only ~20 human mucosal cases have been sequenced, revealing a genomic landscape that is associated with a low single nucleotide mutation burden and no evidence of a UV signature, but numerous large-scale copy number changes and whole-chromosome gains and losses[5–8]. It is also known that the profile of driver genes differs among human melanoma subtypes[6]; for example, BRAF mutations, which are found in around 45% of cutaneous melanomas, are rarely found in mucosal melanoma[5–8]. The abovementioned difference in mutation burden is significant since most of the effort towards developing new therapies for melanoma has shifted to immunotherapies, which have yielded impressive results in tumors with a high mutational load, but appear less effective in mucosal melanomas, which have fewer neo-antigens[9]. Similarly, small molecular drugs that target mutant oncoproteins such as BRAF$^{V600E}$ have been developed, but these agents are not suitable for most cases of the mucosal subtype[10].

Advances in our understanding of cutaneous melanoma have been facilitated by a range of animal models, which have helped to delineate not only the basic biology of the disease, but also the role of driver genes, and the mechanisms of response and resistance to targeted therapies[11–13]. In the context of mucosal melanoma, the most widely accepted animal model is canine melanoma, which bears histological similarities to the human disease and generally develops in the oral mucosa[14]. Dogs have been used extensively to explore the role of new therapies such as vaccination for disease management, but with the exception of analysing driver genes such as BRAF and NRAS, little is known about the genetic landscape of this malignancy and how canine and human mucosal melanomas compare[13,15–17]. Similarly, horses also develop melanoma at sites in or near mucosal tissues, but it is not clear how genetically similar these tumors are when compared to tumors from humans[13]. Intriguingly, equine melanomas are generally more indolent when compared with human or canine mucosal melanomas and as such represent an interesting comparator for a cross-species analysis.

Here, we sequence the exomes of human mucosal, canine oral and equine melanomas and matched normal tissues to characterize the somatic mutational landscape of melanoma in each species, which allowed us to perform a cross-species comparative analysis of recurrent mutations and copy number changes. We also identify germline predisposing alleles implicated in disease development. We find similarities in terms of mutant genes and pathways, suggesting evolutionarily conserved mechanisms of tumor development, but also striking differences between species that help inform on the biology of mucosal melanoma.

## Results

**Sequencing and analysis of melanomas from each species.** We performed whole-exome sequencing of tumor-germline pairs from 46 primary human mucosal melanoma cases, 65 primary oral canine melanoma cases and 28 primary equine melanoma cases. The equine melanomas sequenced originated from mucosal-like (perineum, perianal region, prepuce, vulva, or ventral tail) or mucocutaneous sites (near the eyes or mouth that are in both mucosa and haired skin) (see Methods). For comparison, we also sequenced equine melanomas from cutaneous sites (haired skin only) (32 cases) and other sites (urinary bladder wall muscle and the parotid gland) (2 cases). In addition to primary mucosal cases, we sequenced 5 locoregional recurrences and 8 metastases (7 locoregional and 1 distant) from human patients, and 6 distant metastases from canine patients (Supplementary Data 1). All cases were reviewed by dermatopathologists and medical and/or veterinary pathologists (see Methods). A summary of the cases is provided in Supplementary Data 1, including clinical details such as the sex and age of each patient and time from primary to recurrence or metastasis, where available. We also determined patient ethnicity using a principal component analysis (see Methods and Supplementary Fig. 1). To generate sequencing libraries for the equine cases, we designed whole-exome capture baits covering all protein-coding transcripts annotated in the equine reference genome EquCab2.0 in Ensembl release 79, and for the canine cases we used baits designed using transcripts in the canine reference genome CanFam3.1 in Ensembl release 74 (see Methods). Sequencing libraries for human cases were generated using the Agilent SureSelect All Exon V5 platform. Sequencing was performed to a median depth of 81-, 78-, and 67-fold coverage for human, canine, and equine samples, respectively, after excluding PCR duplicates. To explore the genetic landscape of these melanomas, we generated profiles of somatic point mutations, multi-nucleotide variants, indels, and somatic copy number alterations (SCNAs) using MuTect (v1.1.7)[18], MAC (v1.2)[19], Strelka (v1.0.15)[20], and Sequenza (v2.1.2)[21], respectively, and filtered and quality controlled these variant calls as described in the Methods. Complete lists of the somatic mutations identified in each case are provided in Supplementary Data 2.

**Tumor mutational load and mutational signature analysis.** It was notable that most of the human mucosal, canine oral, and equine melanomas we sequenced had less than 5 mutations/Mb (Fig. 1), unlike UV-associated human subtypes such as superficial spreading and nodular melanoma[6]. We did, however, identify a UV mutation signature (COSMIC signature 7[22], see Methods) in 4 human (3 primaries and 1 locoregional recurrence) and 5 equine samples (Fig. 1 and Supplementary Table 1). The 5 equine tumors, 3 from non-mucocutaneous skin (HD0083a, HD0084a, and HD0071a), one from the vulva (HD0021a) and another from the third eyelid (HD0032a), were located at sun-exposed sites, as was a human sample from the mucosal surface of the lip (PD26932a), while the remaining 3 human samples (PD25657a, PD25643a and matched recurrence PD25643c) were from the nasal mucosa. Notably, a low mutation burden combined with a slight enrichment of C>T mutations has been reported previously in a mucosal melanoma of the nasal cavity in a small sequencing study of 5 samples, suggesting a potential role for UV in the development of some mucosal cases from this site[8]. By pooling all mutations found within each species to identify additional signatures within each cohort, we also identified COSMIC signature 1[22] in the human and canine melanomas, which has previously been found in all cancer types and is known to be correlated with age (see Methods and Supplementary Table 1). No other

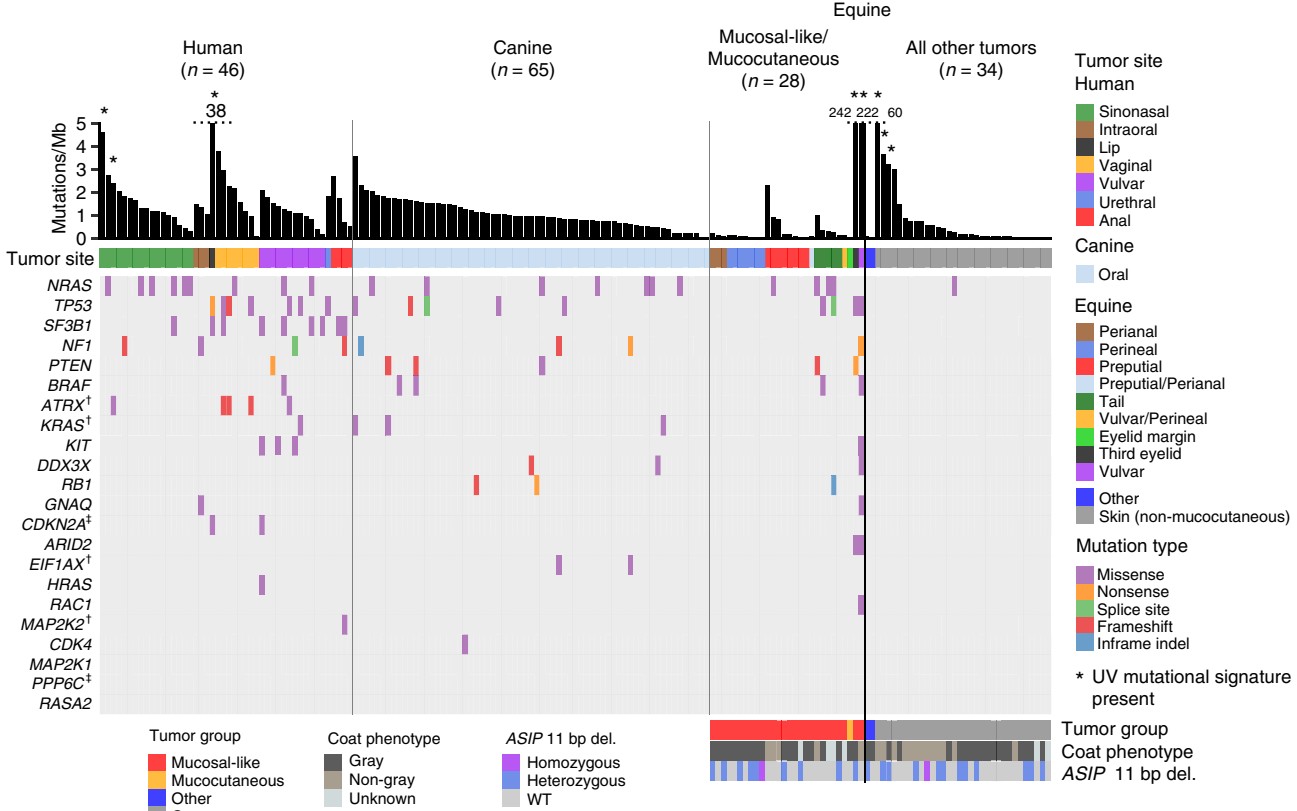

**Fig. 1** Genomic landscape of primary melanomas from human, canine and equine patients. Shown are mutations in established melanoma genes curated from previous studies[6,35], and the exome-wide somatic mutation burden, as mutations/Mb. Mutation type and tumor sites are indicated by color. Equine cases were also classified as mucocutaneous or mucosal-like. As a comparator, primary cutaneous cases and two cases from other sites are also displayed. A full summary of the cases analysed in this study is provided in Supplementary Data 1. For equine cases, gray coat color phenotype is shown, as is the presence of the germline 11 bp *ASIP* deletion in exon 2[70]. Dagger, the gene has not been annotated in the equine genome in Ensembl release 91. Double dagger, the gene has not been annotated in the canine genome in Ensembl release 91

signatures were identified with high confidence. As mutation rates were low in the tumors not affected by UV light exposure, sequencing of additional samples and/or whole-genome sequencing may reveal additional known or novel signatures in mucosal melanoma.

As the gray phenotype, associated with progressive silvering of colored hair, has been linked to melanoma development in horses, we obtained a coat color phenotype for each case from clinical records (Fig. 1 and Supplementary Data 1). In the cutaneous melanomas, samples with a higher mutational load were generally from horses without the gray coat phenotype, suggesting a different path to melanoma development in these cases (Fig. 1). Although the gray phenotype is known to predispose to melanomas in and around mucosal sites, a definitive association between this coat color phenotype and mutational load was not evident for mucocutaneous/mucosal-like cases.

**The driver mutation landscape of melanomas from each species.** Identifying driver mutations and altered pathways allows for the molecular classification of cancers, which can facilitate disease prognostication and management. A recent pan-melanoma study[6] identified 12 significantly mutated genes in a cohort composed of superficial spreading, nodular, acral, and several mucosal melanomas. We first asked which of these genes, or ten other genes previously suggested to be disease associated, were altered in primary tumors from our collection (Fig. 1). In keeping

with previous reports, human mucosal melanomas sequenced in our study carried mutations in genes including *NRAS* (20%; 4 cases with Q61R/K, 2 cases with G12C/V, 2 cases with G13D, 1 case with Y64N), *GNAQ* (2%; V121L), *KIT* (7%; 2 cases with L576P, 1 case with D419H and V654A), *SF3B1* (20%; 6 cases with R625H/C, and 3 other cases), and *TP53* (15%; 8 different mutations in 7 samples), but lacked alterations of several other genes including *CDK4, RAC1,* and *PPP6C,* whose respective mutation has been observed in cutaneous melanoma[5–8]. Also absent were mutations in *POLE, PTCHD2/DISP3,* and *DMXL2,* which were recently reported in a study of Asian patients with mucosal melanoma of the oral cavity[23], while 3 cases (7%) carried *PTPRD* mutations (P823S, T1246M, and V634fs), as reported in the same study (Supplementary Data 2). This discrepancy may reflect differences in ascertainment, as our study presents cases from a range of tissue sites collected from predominantly European-descent patients (Supplementary Fig. 1), but it may also illustrate the heterogeneity of the disease. Intriguingly, while *NRAS* mutations were the most common somatic change in sinonasal tumors, these tumors were devoid of *TP53* mutations, which were common in tumors from other mucosal sites (Fig. 1). Similarly, *SF3B1* R625 mutations were rare in sinonasal tumors, with the majority found in vaginal, vulvar, or anal tumors, as reported previously[24]. The same study reported co-mutation of *NF1* and *KIT,* which we observed in only one case in our sample collection. To determine whether the apparent differences in somatically mutated genes were significantly associated with tumor tissue site or other clinical parameters, we de-sparsified the human somatic

mutation data into biological pathways using SAMBAR[25], performed a principal components analysis (PCA), and a linear regression between the top 5 principal components (PCs) and the available phenotypic variables (tissue site, tumor group, sex, age of diagnosis, or onset) (see Methods). We visualized the results of these associations in a heatmap (Supplementary Fig. 2), which showed that none of the PCs were associated with the phenotypic variables after correcting for multiple testing, suggesting that tumors arising from different mucosal sites do not represent different subtypes, at least at the level of altered pathways. Future efforts to aggregate datasets to increase the sample size and to incorporate addition features such as methylation profiles may reveal subtype differences that were not evident in the current study.

To examine the mutational landscape of established melanoma genes in canine and equine melanomas, we identified the canine and equine orthologs of human genes by searching for genes by gene symbol and obtaining orthology relationships from the Ensembl database (release 91) (Supplementary Data 3). The melanoma genes in Fig. 1 that were not annotated in either canine or equine genomes are indicated. These genes either had no ortholog or had one or more orthologs but no gene symbol assigned. With the exception of canine *MAP2K2*, all other genes in Fig. 1 had one-to-one orthology with a human counterpart (Supplementary Data 3).

Our analysis of canine oral melanomas, widely considered a model of human mucosal melanoma, revealed no *KIT* mutations, however, we identified *NRAS* (11%; 4 cases with Q61H/R/K, 3 cases with G12A/D), *TP53* (8%; 5 different mutations in 5 cases), *BRAF* (3%; G457A, D582G), and *KRAS* mutations (5%; G12D/V, G13D) (Fig. 1 and Supplementary Data 2). Notably, the canine oral melanomas in our series also lacked *SF3B1* and *GNAQ* mutations, and no mutations in *POLE, PTPRD*, or *DMXL2* were found, suggesting that these canine cancers may not represent a faithful genetic model for the subset of human mucosal melanomas with mutations in these genes. Only one case carried a missense mutation in *PTCHD2/DISP3* (Supplementary Data 2). Surprisingly, only two (3%) canine oral melanoma cases carried a truncating mutation in *PTPRJ*, which was recently suggested to be inactivated in as many as 23% of canine mucosal melanomas[26].

We next asked what driver genes were mutated in equine melanomas from mucocutaneous or mucosal-like sites. The most prominent driver genes were *NRAS* (14%; 3 cases with Q61R, 1 case with G12A) and *TP53* (14%, or 4 cases with 6 mutations; H54Y and P27S, R33W, R125W, and R158W and a deletion, g.50612732_50612735del, affecting a splice acceptor site), while two cases had mutations in *BRAF* (7%; V594E, and both P321L and S56L in vulvar tumor HD0021a). Mutations were also observed in *PTEN* (R202*, Y41fs), *KIT* (S339F), and *RB1* (inframe deletion p.N366_R374delinsK), with 14% of samples carrying a mutation in at least one of these genes (Fig. 1 and Supplementary Data 2). Intriguingly, we also found C>T missense mutations in the *KNSTRN* gene, previously implicated in human cutaneous squamous cell carcinoma[27], in 1 cutaneous melanoma (L27F in HD0004a) and two mucosal-like samples with a high mutation rate and UV signature (L27F in HD0021a, and P28L in HD0032a). In humans, a recurrent C>T UV signature-associated hotspot mutation at a nearby residue (S24F) was shown to disrupt chromatid cohesion[27]. However, because there is weak protein sequence conservation between human and horse in this region, functional studies are required to determine the effect of these mutations on *KNSTRN* gene function in horse. As described above for the human mucosal melanomas, we used the SAMBAR algorithm[25] to look for an association between mutational patterns and tissue site and other variables (Supplementary Fig. 2). No associations were found, suggesting that

equine tumors from different mucosal-like or mucocutaneous sites are very similar.

As noted above, as a comparator to our sequencing of equine melanomas from in and around mucosal sites, we sequenced 32 cases of equine cutaneous melanoma from anatomical sites such as the abdomen, thigh and shoulder (Supplementary Data 1). With the exception of one *NRAS* Q61R mutation, these tumors lacked point or indel mutations in known melanoma driver genes (Fig. 1), suggesting a distinct clinical entity or an alternative path to melanoma development. While most equine melanomas are thought to be indolent, some do metastasize[28] and follow-up studies to correlate mutation profiles with tumor behavior are warranted.

**Analysis of mutated genes within and between species.** Following the analysis of established melanoma genes, we proceeded to explore the catalog of mutated genes in each species. In human mucosal melanoma the most notable novel recurrently mutated gene was the *alpha-thalassemia/mental retardation X-linked* (*ATRX*) gene, which carried three frameshift variants and two missense variants (Fig. 1) in five-independent cases (11% of the total). Mutations in *ATRX* have been linked to alternative telomere lengthening in a range of tumor types[29,30]. Notably, telomere dysregulation has been shown to play an important role in cutaneous melanoma through the identification of genes such as *POT1*[31] and mutations in the promoter of *TERT*[32]. Importantly, *ATRX* mutations were found to co-occur with *TP53* mutations (4 of 5 cases; Fisher's exact test, $P = 0.006$). The association of *ATRX* and *TP53* mutations has been noted in other cancers, particularly gliomas, and has been associated with an altered differentiation status[33]. We also observed a *PTPRJ* truncating mutation and two missense mutations, notable because of the aforementioned truncating *PTPRJ* mutations in canine oral melanomas[26] (Fig. 2). Similarly, mutations in *BRCA2* were also found in human and canine cases, but not equine cases.

In addition to mutations in established melanoma genes described above, 2 canine cases were also found to harbor mutations (G9V and R13C) in the highly conserved N-terminal region of the EIF1AX protein (Fig. 1), an essential translation initiation factor that is known to be mutated in human uveal melanoma[34], meningeal melanocytic tumors[35] and other cancers. Indeed, the same amino acid substitutions at conserved sites G9 and R13 have been found in multiple human tumors (COSM6908971, COSM5899335), along with several other recurrent mutations in the N-terminal region, which are predicted to be activating[36]. This conservation strongly suggests that the same mutations in the canine cases play a functional role in these cancers.

In equine melanoma cases from mucosal-like or mucocutaneous sites, the landscape of driver genes was less populated than the other species (Fig. 1), and there were fewer recurrently mutated genes in common with human or canine cases (Fig. 2). In addition to *NRAS* and *TP53*, which were mutated in all 3 species as described above, *PTEN* was disrupted by nonsense or frameshift mutations in all species, suggesting a key role for PI3K signaling in the genesis of these melanomas. Mutations in *RB1* and *FAT3* were found only in canine and equine cases.

In an attempt to identify potential new drivers, we examined the most frequently mutated genes in samples from 2 or more species (Fig. 2), noting genes such as *STAT3* and *TCF7L1* that were mutated in all species, suggesting a potential role in disease development. Mutations in *FAT4* were also seen across species, however, the *FAT4* mutations in the equine samples only occurred in two samples from the vulva (HD0021a) and third eyelid (HD0032a), both of which had a high mutation rate and

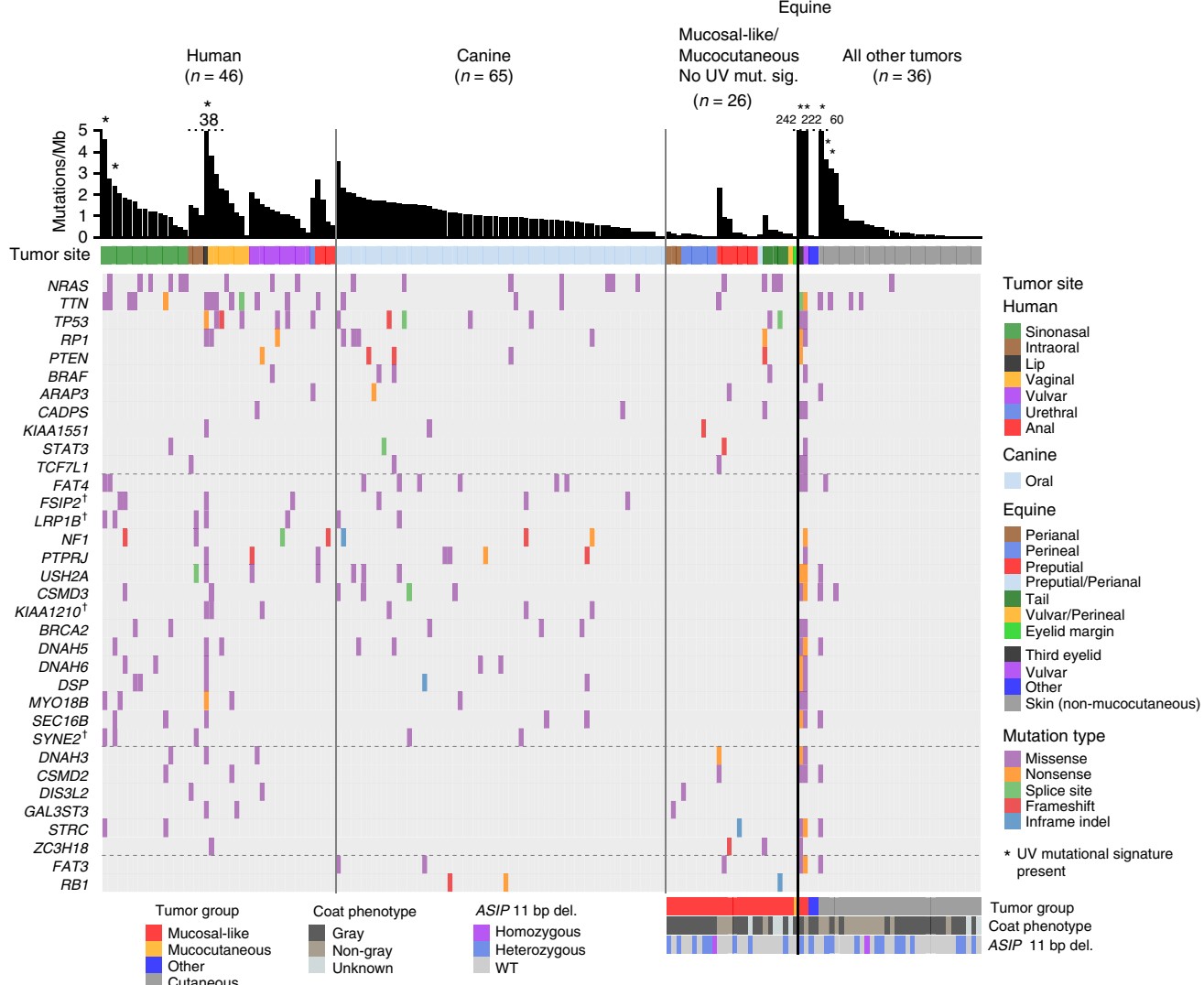

**Fig. 2** Comparative mutational landscape of primary melanomas from human, canine, and equine patients. Shown are the most frequently mutated genes in at least 2 of the 3 species. For genes commonly mutated in human and canine samples only, genes displayed were mutated in at least 5 samples in total, and in at least 3 samples in all other comparisons. When counting mutations and genes, we excluded two equine samples from mucosal-like sites with a high mutation rate and a UV mutation signature, and samples not classified as originating from mucocutaneous or mucosal-like sites; these samples, indicated as "All other tumors", are shown for comparative purposes. Details of tumor site and type are the same as described in Fig. 1. Dagger, The gene has not been annotated in the equine genome in Ensembl release 91

UV mutation signature. *FAT4* has previously been suggested to be a somatically mutated cutaneous melanoma driver gene[37]. All annotated canine and equine genes in Fig. 2 have a one-to-one orthology relationship with the corresponding human gene, with the exception of *DNAH5* (Supplementary Data 3).

**Analysis of somatic copy number alterations.** We next analysed the copy number profiles of human, canine, and equine melanomas. The frequencies of SCNAs in each species cohort is shown in Fig. 3. As reported previously[38,39], human mucosal melanomas and canine oral melanomas harbor extensive SCNAs. Representative copy number profiles of individual samples from each species are shown in Fig. 4. We identified recurrent large and whole-chromosome gains and losses in human mucosal melanoma that mirrored the copy number landscape reported previously using array-based comparative genome hybridization (aCGH)[38], including recurrent gains of 1q, 6p, 8q, 7, and loss of 6q and 10 (Fig. 3a). Similarly, our

canine oral melanomas showed substantial chromosomal gains and losses, most notably, recurrent gains of chromosomes 13 and 17, and losses of chromosomes 2 and 22 (Fig. 3b), which have all been reported previously in canine oral melanoma using aCGH[39]. We also observed a recurrent distinctive complex gain/loss region on chromosome 30, which was previously seen in canine oral melanoma but not in canine cutaneous melanoma[39]. In contrast to human and canine mucosal melanomas, equine melanomas from mucocutaneous/mucosal-like sites had far fewer SCNAs. However, we did observe recurrent whole-chromosome gain of chromosome 25 and loss of chromosome 31, which have not been reported previously, in 50% and 31% of cases, respectively (Fig. 3c). All samples with a copy number loss of chromosome 31 also had a copy number gain of chromosome 25, and the majority of these were perineal or preputial melanomas. This pattern was infrequent in equine cutaneous melanomas, as only 6% of the samples had a loss of chromosome 31 and 3% had a gain of chromosome 25 (Fig. 3c).

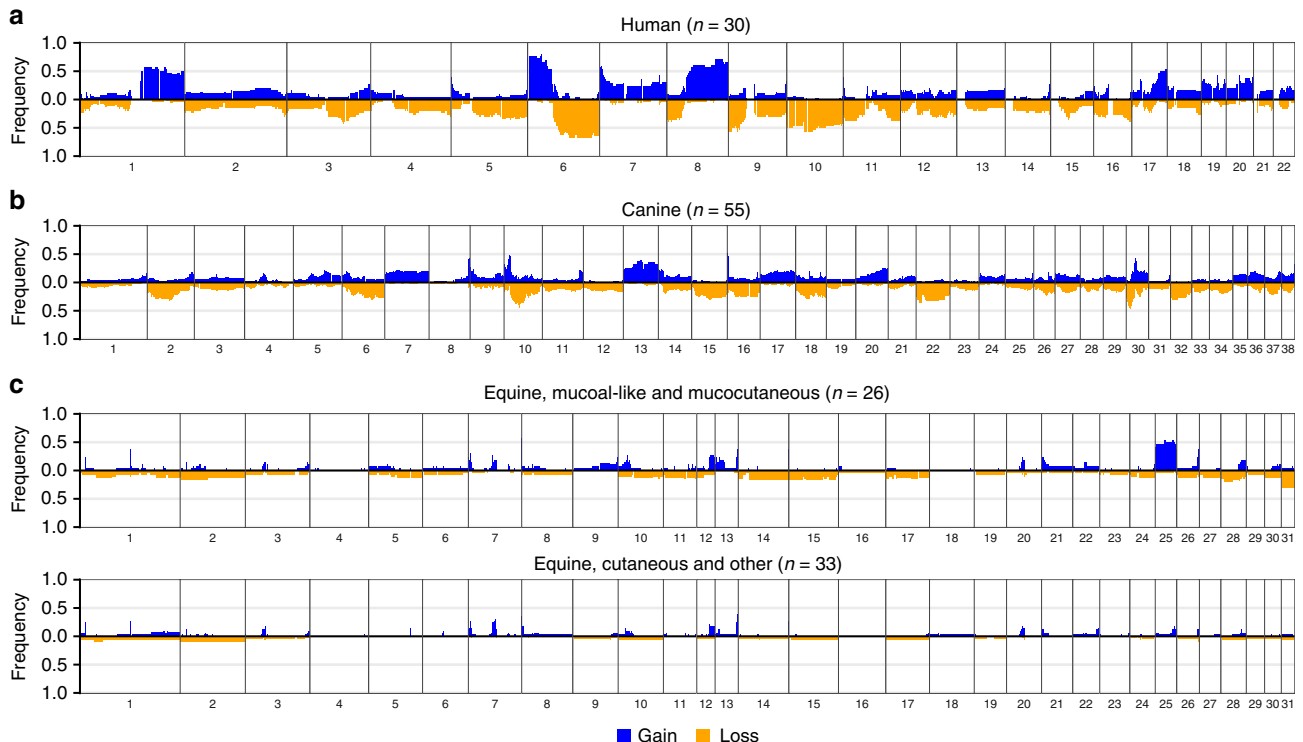

**Fig. 3** The DNA copy number alteration landscape of mucosal melanomas from human, canine, and equine patients. Shown are genome-wide frequencies of somatic copy number gains and losses in primary **a** human mucosal, **b** canine oral, and **c** equine melanomas from mucosal-like/mucocutaneous and cutaneous sites in 1 Mb windows. The pattern of genome-wide frequencies of SCNAs in canine and human melanomas was similar to those reported previously. For equine cases, frequent gain of chromosome 25 and loss of chromosome 31 was observed in tumors from mucosal-like or mucocutaneous sites. Included were samples that passed manual quality control (Supplementary Data 4), as described in the Methods

Candidate driver genes on these chromosomes include *NOTCH1* on chromosome 25 and *ARID1B* on chromosome 31. For visual comparison of copy number frequency profiles relative to human mucosal melanoma, we mapped the frequency of gain and loss in orthologous regions in the canine and equine genomes to the human reference genome (see Methods and Supplementary Fig. 3). Several shared recurrent copy number alterations can be visualized, including a focal deletion on chromosome 15 (human and canine), amplification of the distal end of chromosome 8 (human and canine), partial deletion of 12q (human, canine and equine), and deletion of the distal end of chromosome 6 (human and equine). A detailed cross-species examination of syntenic regions and significant copy number alterations, including these regions, is discussed below.

**Cross-species comparative copy number analysis.** Cancers frequently harbor SCNAs, the majority of which are likely to be passenger events. To identify candidate driver events and genes, we focused on syntenic regions between species where recurrent copy number changes were observed in both species and in the same direction. We used two approaches, first comparing larger SCNAs with a minimal frequency cutoff, and second, using statistical approaches to find significant focal aberrations.

For the first approach, we selected for comparison chromosomes (canine and equine) or chromosome arms (human) where the median frequency of copy number change across the chromosome/arm was at least 0.2 (Supplementary Data 4). Figure 5 shows regions of synteny in these chromosomes/arms, and the location of established and candidate melanoma genes that fall within these regions. The majority of the cross-species recurrent SCNAs are not shared with equine mucosal-like/mucocutaneous melanoma, which

had significantly fewer recurrent SCNAs. Of note, however, there is synteny between equine chromosome 31 and the distal end of human chromosome 6, both of which are recurrently deleted, where *ARID1B* is located in each genome. Equine chromosome 25, where *PPP6C* and *NOTCH1* are located, is syntenic with the distal end of human chromosome 9, however, the recurrent copy number changes in these regions are in the opposite direction. *MYC* is located on syntenic regions in human 8q and canine chromosome 13, both of which are frequently amplified (Fig. 5). Amplification of *MYC* has previously been associated with advanced cutaneous melanoma[40].

In the second approach, GISTIC 2.0[41] and STAC[42] were used to identify significant focal aberrations in each species, which were then compared (see Methods and Supplementary Data 5 and 6). In this way, we identified several genes found in the Cancer Gene Census (CGC) catalog that were focally amplified or deleted in both human and canine mucosal melanoma (Table 1 and Fig. 5). The most significant deletion from the GISTIC 2.0 analysis of human mucosal melanoma was a deletion at 15q15.1 (adjusted *P*-value = 0.009) (Supplementary Data 5), a region with synteny to a significant deletion (STAC footprint-based *P*-value = 0.04) on canine chromosome 30 spanning positions 2–12 Mb (Supplementary Data 6). Within these regions, in both species, are the genes *BUB1B*, *KNSTRN*, and *B2M*. *B2M* is frequently altered in human tumors and plays a pivotal role in immune evasion[43], and both *KNSTRN* and *BUB1B* are involved in chromosome segregation[27,44]. The conserved copy number alteration of these genes suggests they represent strong candidate driver genes. This region also contains *SPRED1*, a gene recently shown to be a mucosal melanoma driver[45]. We note that Poorman et al.[39] previously identified a significant focal deletion at ~7.2–7.7 Mb (relative to CanFam3.1) on chromosome

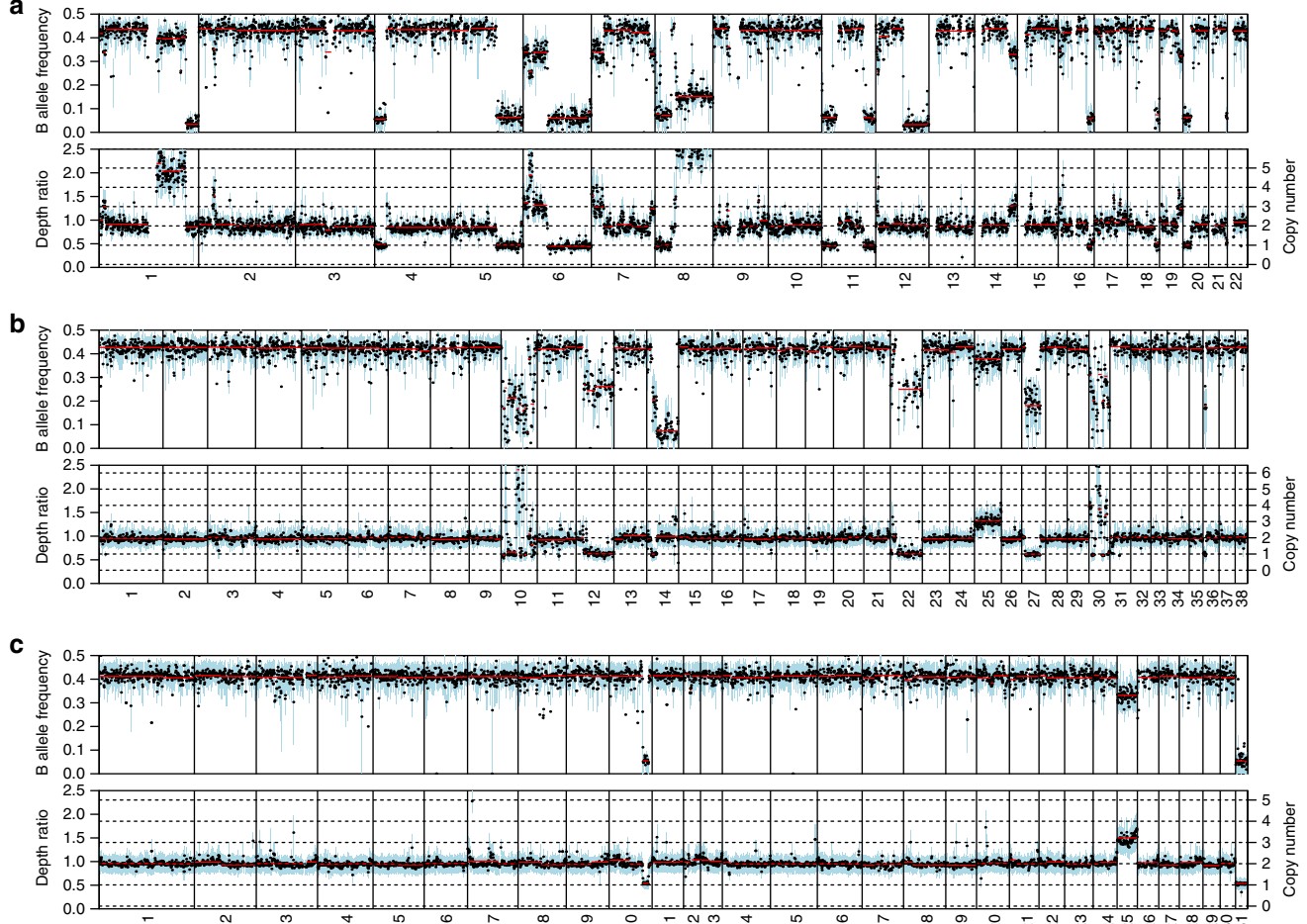

**Fig. 4** Copy number profiles of individual melanoma samples. **a** A representative copy number profile of a human mucosal melanoma from the nasal sinus (PD26997a). This case had copy number gains of 1q, 6p and 8q, and loss of 6q, 8p, all of which are known recurrent events in mucosal melanoma[38]. **b** A copy number profile of a representative canine oral melanoma (DD0068a), with complex copy number profiles on chromosomes 10 and 30, loss of chromosomes 12, 22, 27, and gain of chromosome 25. **c** A representative copy number profile of an equine mucosal-like melanoma from the prepuce (HD0081a). This sample harbors both a gain of chromosome 25 and loss of chromosome 31, which were frequently observed (although not necessarily together) in the equine cases from mucosal-like sites

30 in canine oral melanomas; this smaller region contains *KNSTRN* and *BUB1B* but not *B2M*. In the equine genome, these genes are located in a syntenic region on chromosome 1 (Fig. 5) where no recurrent SCNA was observed. However, as noted above, missense mutations in *KNSTRN* were observed in 2 mucosal-like and 1 cutaneous equine sample.

Cross-species comparison of regions with significant copy number gain revealed that the mouse double minute 2 homolog (*MDM2*) gene is focally amplified in human and canine mucosal melanomas (Table 1). *MDM2* is a well-established oncogene in human cutaneous melanoma and functions to antagonize the activity of *TP53*[46]. Also of note, was the amplification of *SMO*. *SMO* is a well-established oncogene in a range of cancers including medulloblastoma[47] and its amplification in both human and canine mucosal cases suggests a functional role in tumorigenesis (Table 1). The orthology relationships of genes shown in Table 1 and Fig. 5 are listed in Supplementary Data 3. The conserved partial deletion on human 12q, described above, maps to syntenic regions on equine chromosome 28 and canine chromosomes 10 and 15 that contain Cancer Gene Census genes *BTG1*, *CHST11*, and *USP44*. However, no significant focal deletions of these regions were identified by STAC analysis.

**Comparison of primaries to metastatic and recurrent tumors.** Little is known about tumor heterogeneity when it comes to mucosal melanoma. To address this question, we sequenced eight metastatic human mucosal melanomas from seven cases (1 distant, 7 locoregional), which were paired with primary tumors included in the analyses described above (Supplementary Data 1). We sequenced an additional 3 locoregional recurrences that had developed at the primary site following surgical removal of the primary, and for two of the cases above with primary and matched metastases, we also sequenced locoregional recurrences. In addition, we sequenced six metastatic canine oral melanomas (all distant metastases) from five patients (Supplementary Data 1). Somatic mutations in the recurrences and metastases were identified relative to the matched normal samples, and compared to mutations in their corresponding matched primary sample. (Fig. 6). The greatest overlap of somatic mutations in a metastasis was 59% for human cases (median 42%), and 71% for canine cases (median 44%). This suggests that there is either significant heterogeneity in the primary tumor or evolution of the metastasis once it has left the primary site. In support of the heterogeneity model, recurrences shared as many mutations with primaries as metastases (23–61%; median 34%), suggesting that mucosal

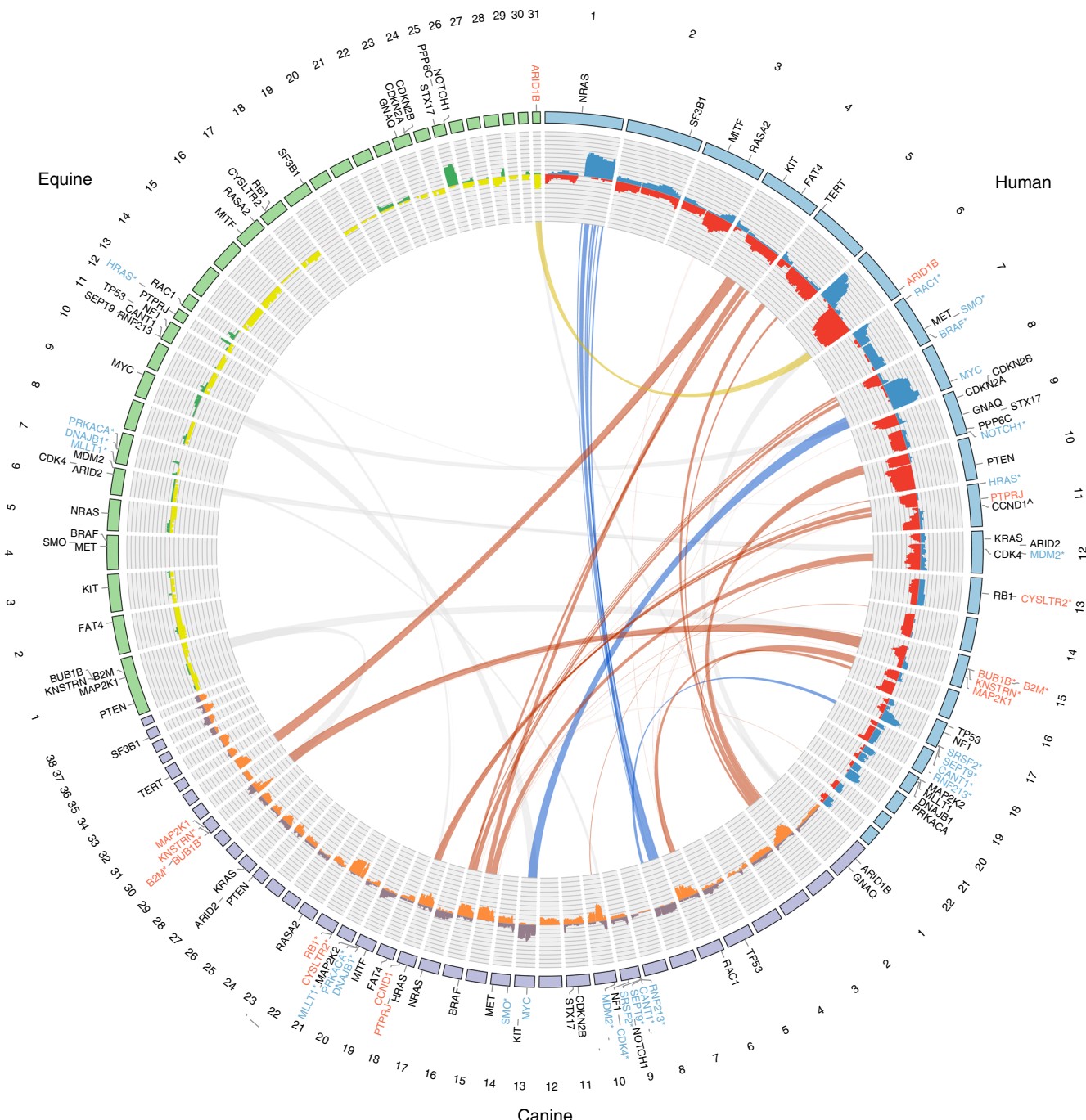

**Fig. 5** Cross-species comparison of somatic copy number alterations in melanoma. Equine cases used were from mucosal-like or mucocutaneous sites. Chromosomes are represented by the outer track. Genes shown are established or candidate melanoma genes. The histogram shows the frequencies of copy number (CN) gains (blue, purple, green) and losses (red, orange, yellow) in each species. The blue ribbons indicate regions of synteny on canine chromosomes and human chromosome arms that have a median frequency of CN gain of at least 0.2 in both species. Similarly, the orange and yellow ribbons indicate syntenic regions on canine and human chromosomes/arms, and on equine and human chromosomes/arms, respectively, with a median frequency of CN loss of at least 0.2 in both species. Gray ribbons show syntenic regions where the frequency of CN gain or loss was below 0.2. Genes shown in blue or red with an asterisk (*) are located within a significant focal CN gain or loss, respectively. The remaining genes in blue or red are located on chromosomes/arms in syntenic regions linked by blue or orange/yellow ribbons, respectively. ^CCND1 is on human chromosome 11, which had a frequency of CN loss>0.2, however, this gene is in focally amplified region from GISTIC 2.0 analysis with q-value = 0.11 (Supplementary Data 5)

melanomas from both humans and dogs are composed of multiple subclones. This supports the notion that after tumor initiation there is appreciable evolution of the primary tumor[48], which may potentially confound attempts at disease management. This question should be explored further with ultra high-depth sequencing of multiple tumor regions.

**Analysis of sequence data for pathogenic germline variants.** In addition to our analysis of somatic mutations, we also looked for pathogenic germline variants in established human melanoma susceptibility genes[49], including *CDKN2A*, *BAP1*, *POT1* and *TP53*, and their orthologs in the canine and equine genomes. In the human cases we found one patient (with multiple germline

**Table 1 Cross-species comparison of candidate genes in regions of significant copy number gain or loss**

| Region limits (Human) | CGC genes | Region limits (Canine) | CGC genes | Region limits (Equine) | CGC genes |
|---|---|---|---|---|---|
| *CN gain* | | | | | |
| 12:68.2–71.6 Mb | **MDM2\***, *PTPRB\** | 10:1.0–12.0 Mb (10:6.5–12.0 Mb) | *NAB2, STAT6, GLI1, DDIT3, CDK4, LRIG3, WIF1\**, **MDM2\*** | | |
| 17:80.1–82.1 Mb/ 17:63.5–83.2 Mb (17:71.7–82.3 Mb) | *CD79B, DDX5, AXIN2, PRKAR1A, H3F3B\**, **SRSF2\***, **SEPT9\***, **CANT1\***, **RNF213\***¹, *ASPSCR1\** | 9:1.0–6.0 Mb | **RNF213\***, **CANT1\***, **SEPT9\***, **SRSF2\*** | | |
| 7:124.1–143.1 Mb (7:128.6–134.6 Mb) | *POT1, SND1,* **SMO\***, *CREB3L2, TRIM24, KIAA1549, BRAF* | 14:1–8.0 Mb (14:2.9–7.9 Mb) | **SMO\*** | | |
| | | 20:48.0–55.0 Mb (20:48.0–52.0 Mb, 20:53.3–55.0 Mb) | **DNAJB1\***, **PRKACA\***, *CALR\*, SMARCA4\*, DNM2\*, KEAP1\*, CD209, VAV1\**, **MLLT1\*** | 7:3.0–4.0 Mb, 7:44.0–45.0 Mb (7:44.1–45.5 Mb) | **MLLT1\***, **DNAJB1\***, **PRKACA\*** |
| 11:1–2.0 Mb/ 11:1–3.7 Mb (11:0.19–3.0 Mb) | **HRAS\***² | | | 12:24.0–32.0 Mb (12:30.2–32.0 Mb) | *MEN1,* **HRAS\***, *CARS\** |
| *CN loss* | | | | | |
| 15:36.0–44.0 Mb/ 15:32.8–45.4 Mb | *NUTM1\**, **BUB1B\***¹, **KNSTRN\***¹, **B2M\*** | 30:2.0–12.0 Mb | **BUB1B\***, **KNSTRN\***, **B2M\*** | | |
| 13:48.7–69.7 Mb | **CYSLTR2\***³ | 22:1–13.0 Mb | **CYSLTR2\***, *RB1\*, LCP1\** | | |

Region limits of significant copy number alterations in the canine and equine genomes were identified using STAC, and both STAC and GISTIC 2.0 were used for human. Listed are genes found in the Cancer Gene Census (CGC) catalog. Genes in regions with significant copy number gain or loss in at least 2 species are shown in bold. Genomic coordinates in parentheses indicate the sequence within the region limits that are syntenic; otherwise, the whole region is syntenic. If both STAC and GISTIC 2.0 region limits are listed, then the synteny was determined from the larger region. Genomic coordinates are relative to GRCh38 (human), CanFam3.0 (canine) and EquCab2.0 (equine)
\*Gene is within the boundaries of the syntenic region
¹Gene is within STAC and GISTIC 2.0 region limits, otherwise found either within STAC or GISTIC 2.0 region limits (human only)
²HRAS was found within a region both recurrently gained and lost within the human cohort. STAC found both the recurrent copy number gain and loss significant while the recurrent loss was significant with GISTIC 2.0
³CYSLTR2 overlaps the lower boundary of the region limits

samples sequenced, PD25652b/e/f with a pathogenic complex *TP53* mutation (11 bp deletion + 5 bp insertion; ClinVar ID: 12381) which is in keeping with a previous case report outlining the development of mucosal melanoma in a Li Fraumeni syndrome patient (Table 2)[50]. No known pathogenic or risk factor variants (as described in the ClinVar database), or deleterious substitutions (defined by a SIFT score <= 0.05) were identified in *CDKN2A* or *BAP1*, while a single case (PD25655b) carried a *POT1* Q539H variant which has not been reported previously in the ClinVar database but is present in dbSNP (rs973319258) and the gnomAD database with a population variant allele frequency (VAF) of $1.6 \times 10^{-5}$. This variant is predicted to be deleterious (SIFT score = 0.05). Extending our search for pathogenic variants, risk factor variants and deleterious substitutions to all genes, we identified patients with truncating loss-of-function mutations in *BRCA1* (E23fs; ClinVar ID: 17662; PD26992b) and *BRCA2* (L122fs; ClinVar ID: 51504; PD25663b) (Table 2), both of which are associated with predisposition to breast and ovarian cancer. The *BRCA1* E23fs (p.Glu23Valfs) mutation, also known as 185delAG, is a founder mutation in Ashkenazi Jews[51,52]. Prior to our study, *BRCA1* and *BRCA2* have not been associated with mucosal melanoma and thus we extend the list of genes linked to this disease. Since tumors developing in *BRCA1* and *BRCA2* patients may be sensitive to treatment with PARP inhibitors[53], a germline analysis of patients presenting with mucosal melanoma, particularly if their pedigree also contains cases of breast and ovarian cancer, may be warranted. Another interesting observation was a single patient with a pathogenic *MUTYH* Y176C variant (ClinVar ID: 5293). *MUTYH* variants have not been linked to melanoma previously and although *MUTYH* is generally considered a recessive tumor suppressor gene it is known that individuals heterozygous for disruptive *MUTYH* alleles are at greater risk of developing a range of cancers[54]. Finally, we noted a patient with a pathogenic *CHEK2* mutation (p.S471F; also known as p.S428F; ClinVar ID: 5603) which has been linked to a range of cancers and is also thought to be an Ashkenazi founder allele[55].

Thus, we have observed an intriguing concentration of pathogenic germline alleles in this cohort with disruptive mutations found in *TP53, POT1, BRCA1, BRCA2, MUTYH* and *CHEK2*.

None of the canine or equine cases in our study carried potentially pathogenic (SIFT score <= 0.05) germline mutations in the abovementioned genes, with the exception of 2 equine cases with *MUTYH* L241V mutations (SIFT score = 0.01), that developed melanoma on the prepuce (HD0024a) and ventral neck (HD0026a). We have also reported the presence of the gray phenotype, caused by a germline intronic duplication in *STX17* that activates the MAPK pathway[56], and the 11 bp *ASIP* exon 2 deletion (Fig. 1 and Supplementary Data 1), both previously linked to the regulation of coat coloration and a higher incidence of melanoma in horses[56].

**Identification of actionable mutations in mucosal melanoma.** Our catalog of driver mutations revealed several actionable mutations. These included activating mutations in the MAPK pathway (*NRAS, BRAF, NF1,* and *KRAS*) which could be targeted using drugs such as trametinib (Fig. 1)[57]. In all species, we also observed mutations in the PI3K pathway including loss-of-function mutations in *PTEN*, potentially targetable using PI3K inhibitors. In human and canine samples, and two equine samples from mucosal-like sites with a UV mutation signature, we found somatic mutations in *BRCA2*, which might suggest that these tumors would respond to PARP inhibitors. Similarly, germline mutations in both *BRCA1* and *BRCA2* were identified in human patients in our study. Importantly, in human and canine mucosal cases we also found recurrent amplifications of *SMO*, which can be inhibited using several compounds, including Glasdegib, which has recently been approved by the United States Food and Drug Administration. Finally, we note the presence of somatic mutations in other genes for which small molecular inhibitors have been developed, such as *ROS1*. Follow-up functional studies will be required to determine how tumors respond to these agents.

**a**  Primary and metastatic (human)

**b**  Primary and recurrent (human)

**c**  Primary, metastatic and recurrent (human)

**d**  Primary and metastatic (canine)

**Fig. 6** Comparison of somatic mutations in recurrences and metastases to primary tumors. Shown are the number of shared and private mutations in each sample. Blue represents primary tumors, red represents metastatic tumors, and green represents recurrent tumors. Tumor details can be found in Supplementary Data 1

## Discussion

Here we sequenced the exomes of human mucosal, canine oral, and mucosal-like/mucocutaneous equine melanomas and performed a cross-species comparative genomic analysis. We identified similarities and differences in the mutation profiles, both in terms of the mutated driver genes but also in terms of mutation number, which are likely to influence tumor behavior and response to treatments. Further, we reveal divergent DNA copy number profiles, with human and canine mucosal melanomas showing substantial copy number gains and losses, while equine cases appeared to have comparatively fewer copy number changes. Importantly, we identified recurrent SNCAs and, using a cross-species comparative analysis, were able to further refine the loci that may functionally contribute to disease development. Our cross-species analysis of human and canine melanomas is particularly important since canine patients are used as spontaneous models of human mucosal melanoma, and while we found many similarities, such as mutations in *NRAS, TP53*, and *NF1*, the genomes of canine melanomas lacked mutations in other key human mucosal melanoma drivers, such as *SF3B1* and *ATRX*. Thus, dogs may not represent a faithful genetic model for these sub-types of human mucosal melanoma. In this study, we aimed to compare tumors of mucosal origin between species. It is notable that in humans, other forms of melanomas, such as acral lentiginous melanoma[6], may form on sun-shielded skin and the mutation profiles we describe for canine oral and horse mucosal-like/mucocutaneous melanoma bear some similarities to these diseases. Similarly, we show that in horse, melanomas from in or around mucosal sites appear to be different from melanomas from sun exposed cutaneous sites, where the latter may show some similarities to melanocytoma[58]. Collectively, our study details the genetic differences and similarities between mucosal melanoma from human, dog, and melanomas from horses and should help inform studies into the biology of this disease.

## Methods

**Samples and DNA extraction**. For our human cohort, we collected 122 samples total from various tissues (36 primary and matched normal pairs; 10 sets of recurrent and/or metastatic tumors with 1 or more matched normal samples; 5 metastatic and matched normal pairs; 1 recurrence and matched normal pair). We also collected 136 canine samples (60 primary and matched normal pairs, 4 trios of primary, matched normal and metastatic samples, and 1 set of primary, matched normal, and two metastatic samples) and 124 equine samples (62 primary and matched normal pairs). Metastases included both distant and locoregional metastases, the former defined as tumors forming outside the lymphatic drainage region. Human mucosal melanoma cases were obtained from three clinical centers (University of Michigan, University of Edinburgh and University of California, San Francisco) and were reviewed by specialist dermatopathologists. All cases were ethically approved for this study by local Institutional Review Boards and by the Sanger Institute's human materials and data management committee. The equine melanomas were acquired from four large veterinary pathology practices (Rossdales Equine Hospital, the Animal Health Laboratory and the Department of Pathobiology at the University of Guelph, University of Edinburgh and University of Cambridge) and were reviewed by a team of medical and veterinary pathologists. Canine cases (from the Animal Health Laboratory and the Department of

**Table 2 Selected germline variants in human mucosal melanomas**

| Gene | Chr. | Position | Ref. | Alt. | Variant effect | Prot. Pos. | AA change | SIFT score | dbSNP ID/ COSMIC ID | ClinVar clinical significance (Variation ID; Alt. name) | Case(s) |
|---|---|---|---|---|---|---|---|---|---|---|---|
| *BRCA1* | 17 | 43124027 | ACT | A | Frameshift | 23 | E/X | — | rs386833395 COSM3190163 | Pathogenic (17662; p.Glu23fs; c.68_69 delAG; 185delAG) | PD26992b |
| *BRCA2* | 13 | 32338034 | CTG | C | Frameshift | 1227 | L/X | — | rs80359395 | Pathogenic (51504) | PD25663b |
| *TP53* | 17 | 7676037 | CAGACGGAAACC | CTGAAT | Inframe deletion (11 bp del + 5 bp ins | 108-111 | GFRL/ IQ | — | — | Pathogenic (12381) | PD25652b/e/f |
| *MUTYH* | 1 | 45332803 | T | C | Missense | 176 | Y/C | 0 | rs34612342 | Pathogenic (5293; p.Tyr179Cys) | PD26990b |
| *CHEK2* | 22 | 28695219 | G | A | Missense | 471 | S/F | 0.01 | rs137853011 COSM2935967 | Conflicting interpretations of pathogenicity; Risk factor (5603; p.Ser428Phe) | PD26997b |
| *MITF* | 3 | 69964940 | G | A | Missense | 419 | E/K | 0.04 | rs149617956 | Conflicting interpretations of pathogenicity; Risk factor (29792; p.Glu318Lys) | PD25648b/e |
| *TMEM127* | 2 | 96265174 | C | T | Missense | 70 | D/N | 0.36 | rs121908819 | Conflicting interpretations of pathogenicity (126964) | PD26692b |
| *ATM* | 11 | 108227849 | C | G | Missense | 49 | S/C | 0 | rs1800054 | Benign/Likely benign; Risk factor (3048) | PD26932b PD26998b |
| *COL6A3* | 2 | 237378831 | G | A | Missense | 768 | R/C | 0 | rs200722892 (G > C) | — | PD26923b |
| *POT1* | 7 | 124827283 | T | A | Missense | 539 | Q/H | 0.05 | rs973319258 | — | PD25655b |

Shown are germline variants in human mucosal cases from this study that were found in the ClinVar database and classified as a risk factor for cancer, had a clinical significance of either pathogenic or mixed pathogenicity, or had a deleterious SIFT score <= 0.05. "Ref." and "Alt." are the reference and alternate bases, respectively, at each position. "AA change" is the amino acid substitution resulting from the alternate base. Identifiers from dbSNP, and the COSMIC and ClinVar databases are shown. Note that variants in the COSMIC database are somatic and ClinVar variants shown were observed previously as germline variants

Pathobiology at the University of Guelph and the University of Edinburgh) were reviewed by three-independent veterinary pathologists. Equine melanoma cases were further classified as originating from mucocutaneous or mucosal-like sites, the most common being from the prepuce and perineum, or cutaneous sites (see "Classification of equine tumors" below). The cases from mucosal-like or muco-cutaneous sites were used for comparison to the human and canine melanomas, while the equine cutaneous melanomas were primarily used as a comparator to the equine mucosal-like and mucocutaneous melanomas. All samples and clinical details are listed in Supplementary Data 1. All samples were obtained as paraffin-embedded tissue cores, and DNA was extracted with the QIAamp FFPE Tissue kit from Qiagen, according to the manufacturer's instructions.

**Classification of equine tumors**. In order to compare equine melanomas to the human mucosal and canine oral melanomas, we classified the equine tumors as cutaneous (tumors arising on haired skin only), mucocutaneous (tumors near the eyes or mouth that are in both mucosa and haired skin), and mucosal-like (tumors arising on the perineum, perianal region, prepuce, vulva, or ventral tail). Two tumors, from urinary bladder wall muscle and the parotid gland, did not fall into any of these 3 categories, and were classified as 'other'. Samples and their classifications are listed in Supplementary Data 1.

**Sequencing and variant calling**. Exome capture was performed using custom designed and Agilent SureSelect baits as described in the Results. Paired-end sequencing was performed using the Illumia HiSeq platform at the Wellcome Trust Sanger Institute to generate 75 bp reads. Sequencing reads were aligned using BWA-MEM (v0.7.12)[59] to reference genomes GRCh38, CanFam3.1, and Equ-Cab2.0 for human, canine, and equine samples, respectively. PCR duplicates, secondary read alignments, and reads that failed Illumina chastity (purity) filtering were flagged and removed prior to running variant and copy number calling. Also removed were read pairs where one read had the same sequence as another read, but their mates had mapping quality 0 and mapped to different sites. These reads appeared to be artefacts and frequently carried sequencing errors which confounded variant calling. The resulting sequencing coverage ranged from 25- to 115-fold (median 81-fold) for the human samples, 34- to 124-fold (median 78-fold) for the canine samples, and 24- to 131-fold (median 67-fold) for the equine samples. MuTect (v1.1.7)[18] and Strelka (v1.0.15)[20] were used to call somatic SNVs and indels, respectively. Germline variants were identified using the Genome Analysis Tool Kit (v3.5) HaplotypeCaller followed by joint genotyping[60]. The minimum base quality score for somatic and germline variant calling was set to Phred 30. The list of SNVs from MuTect were input into MAC (v1.2)[19] to identify multi-nucleotide variants, such as UV-induced CC > TT substitutions. The Ensembl Variant Effect Predictor[61] was used to predict the effect of variants on genes and

proteins, relative to Ensembl version 89 for human samples and Ensembl version 91 for canine and equine samples. To remove artefacts, MuTect variant calls were removed if any of the following criteria were met: (1) total read depth at the position <10 in the tumor; (2) the VAF < 0.05; (3) total number of reads with the variant allele ≤5 in the tumor, and VAF < 0.15; (4) total read depth in the tumor <20 and VAF < 0.2; (5) the matched normal sample had 3 or more reads with the variant base. Variant calls were also removed if the variant base was observed in 3 or more unrelated normal samples or tumor samples in which a somatic variant call was not made. Human variant calls found in the ExAc or gnomAD databases[62] were removed if the population variant allele frequency was greater than 0.01. Known common germline variant calls found in canine dbSNP version 151, DogSD[63] and a cohort of 505 recently sequenced dogs were removed from the list of canine somatic variant calls. For the equine somatic variant call set, variants found in dbSNP 151 were removed. The alignments for all variants reported in Table 2, Fig. 1 and Fig. 2 were visually inspected. In human, dog, and horse samples, variants found in *MECOM*, *KDM3B*, and *BAP1*, respectively, were removed as they appeared to be artefactual calls due to alignment of contaminating cDNA reads.

**Ethnicity analysis**. To infer the ancestry of mucosal melanoma patients, germline variant calls from these individuals were intersected with variants from 1092 individuals from the 1000 Genomes Project Phase I (1KGP)[64]. Variants taken forward for principal component analysis (PCA) included high quality variants in the melanoma cohort (as described above) that were also found in 1KGP at a minor SNP allele frequency >= 0.05, that were not in linkage disequilibrium with another SNP (pairwise $R^2$ < 0.02), and that did not have a missing rate >0.05. After filtering, 5804 SNPs remained that were spread across all autosomes. The first ten principal components were estimated using all individuals together. All analyses were carried out with the SNPRelate R package[65].

**Extraction of mutational signatures**. The SomaticSignatures R package[66] was used to extract mutational signatures from species cohorts and determine the proportion of mutations in each sample attributable to specific signatures. To identify signatures within each species cohort, samples within each species were divided into 2 groups and mutations were pooled within each group. To maximize the input available for the algorithm, all primary, recurrent and metastatic samples were also included, if available, and all equine cutaneous samples and samples from the urinary bladder wall muscle and parotid gland. The human samples were grouped by primary tumor site (sinonasal and oral tumors in one group, anogenital and vulvar in the other), and the equine samples were grouped by tumor type (mucosal-like/mucocutaneous in one group, and cutaneous, urinary bladder wall muscle and the parotid gland tumors in the second group). Since the canine samples were all oral melanomas, we arbitrarily grouped the samples into 2 groups (35 and 36 samples) to perform a signature analysis for the canine cohort. In the human cohort, the optimal solution contained 3 signatures, which were compared to the 30 COSMIC signatures identified by Alexandrov et al[67] using cosine similarity. Two of the extracted signatures corresponded to signature 7 (cosine similarity = 0.96), which is associated with UV light exposure, and signature 1 (cosine similarity = 0.80), which is seen in all cancer types and associated with age. The third signature only had weak cosine similarity to signatures 5 (cosine similarity = 0.74) and signature 30 (cosine similarity = 0.75), which we did not consider significant matches. Signature 1 was extracted from the canine cohort (cosine similarity = 0.87). Signature 7 was extracted from the both the mucosal and cutaneous equine cohorts (cosine similarities = 0.96 and 0.94, respectively).

To assign mutation signatures to specific samples, SomaticSignatures was run on species cohorts without pooling mutations. Only samples with at least 100 mutations were included, as those with fewer mutations would be uninformative. In total, 11 human samples (including 8 primaries, 2 metastases, and 1 recurrence) and 5 equine primaries were analysed. Only 1 canine sample had more than 100 somatic SNVs, therefore, mutational signature analysis was not applicable. As with the pooling approach described above, a signature with highest cosine similarity to signature 7 (0.88) was identified in the human cohort. Signatures with highest similarity to signature 5 (0.79) and signature 1 (0.68) were identified, however, using visual inspection of mutation profiles we were not able to determine whether the extracted signatures truly represented signatures 1 and 5, due to the sparseness of the data. Signature 7 (cosine similarity = 0.96) was identified in 5 samples in the equine cohort. Samples in which signature 7, a signature associated with UV light exposure, comprised at least 50% of the total signature contribution are indicated in Figs. 1 and 2.

**Comparison of mutated pathways across tumor sites**. To compare mutational patterns across mucosal melanoma tumor sites, we used Subtyping Agglomerated Mutations By Annotation Relations (SAMBAR)[25]. SAMBAR's input is a matrix containing the number of mutations in each gene and each sample. SAMBAR normalizes the number of mutations to the gene's length and de-sparsifies these gene mutation scores into biological pathway mutation scores, which can be used for subtyping analysis or to identify associations between mutational patterns and phenotypic variables. For each human and equine gene, gene length was calculated from the total coding sequence (CDS) length of the canonical transcript, as defined

by Ensembl release 89 for human genes and release 91 for equine genes. For the analysis on equine samples, we only included genes with gene symbols and a one-to-one homology to human. We used the binomial dissimilarity index on the pathway mutation scores we obtained from SAMBAR to calculate distances between samples within each species dataset. Next, we regressed the top five principal components from the PCA on these distance matrices with each phenotypic variable. We adjusted the P-values of the regression for multiple testing using Benjamini and Hochberg's method across each variable and principal component (Supplementary Fig. 2).

**DNA somatic copy number alteration calling**. Sequenza (v2.1.2)[21] was used to estimate tumor cellularity and ploidy from paired tumor-normal WES data, and calculate allele-specific copy number profiles. For each sample, the best Sequenza solution was chosen after visual inspection of both the best-fit solution (with the maximum log posterior probability) and alternative solutions. We observed several cases where all solutions provided estimates of cellularity that were less than 0.2. Upon visual inspection, however, we discovered high log posterior probabilities for alternate solutions with estimates of ploidy 2 and cellularities spanning 0–1. Based on this, and our expectation that the tumors have minimal contamination from normal tissue, we assigned a cellularity of 0.8 and ploidy 2 for the calculation of copy number profiles for these samples. Samples with excessive noise and no apparent optimal solution (16 human, 10 canine, 2 equine mucosal, and 1 equine cutaneous) were excluded. Supplementary Data 4 lists the samples used for copy number analysis. For the canine and equine samples, segments were further filtered by removing segments with altered copy number if they were less than 5 Mb in size and overlapped 5 or more reference genome assembly gaps larger than 1 kb. Genomic regions near multiple assembly gaps tend to be repetitive in nature; both gaps and repetitive sequence can result in spurious read mapping and, in turn, incorrect copy number calls from Sequenza. As the human reference genome is much more refined than the canine and equine genomes, this filtering was not necessary for the human data.

**Humanization of canine and equine copy number data**. For visual comparison of canine and equine mucosal melanoma copy number frequency profiles to that of human, we used a method similar to Poorman et al.[39] to identify orthologous regions. Genome coordinates of the 60-mer oligo probes from the Agilent SurePrint G3 Human CGH 4x180 Microarray (provided relative to reference genome hg18/GRCh36) were mapped to the human reference genome GRCh38 using the liftOver utility. The frequency of copy number gains and losses of each re-mapped region was then plotted against the genomic position. Each probe region was then mapped, using liftOver, to the orthologous region in CanFam3.1 to determine the frequencies of gains and losses in the canine cohort, which were plotted relative to GRCh38. This was repeated for the equine copy number data, by mapping the human probe regions to EquCab2.0.

**Cross-species comparative analysis of recurrent SCNAs**. In each species, we first identified whole chromosomes (canine and equine) and chromosome arms (human) with recurrent SCNAs by calculating the median frequency of copy number gains across chromosomes/arms using 1 Mb windows. This was repeated for copy number losses. Chromosomes/arms with a minimum copy number frequency 0.2 were used for cross-species analysis. Syntenic regions between the human, canine, and equine genomes were retrieved from the Ensembl Compara database[68]. The syntenic regions were overlapped with the chromosomes/arms with recurrent gain or loss, and candidate and established melanoma genes within these overlap regions were identified.

To identify significant focal SCNAs in the human melanoma cohort, we used GISTIC 2.0 (v2.0.23)[41] and STAC (v1.2)[42]. To transform the results from Sequenza (described above) into GISTIC 2.0 input, segment chromosome, start and end positions were used, and the segment log-scaled copy number ratio was calculated as $\log_2(2 \times \text{depth-ratio})-1$. The optional markers file was not used and default parameters were selected. Peaks with both q-value and residual q-value <0.05 were selected as significant. For each significant wide peak region and region limit, genes found in the Cancer Gene Census (CGC)[69] catalog were identified. For STAC, regions of copy number gain and loss from Sequenza were formatted separately in 'location' format using 1 Mb spans. Each chromosome arm was analysed separately. A region with a minimum copy number frequency of 0.2 and a frequency-based P-value <0.05 or footprint-based P-value <0.05 was considered significant. Because the current release of the GISTIC 2.0 pipeline was designed for the analysis of human cancers, we used STAC for the analysis of SCNAs in canine and equine mucosal melanoma, as the algorithm is not species-specific. Whole chromosomes were analysed individually. The same input format and filtering used for human SCNAs were applied. For each significant region, CGC genes were identified. Samples included in GISTIC 2.0 and STAC analysis are listed in Supplementary Data 4, excluding the equine samples listed as "cutaneous and other".

For the human cohort, each significant region with a CGC gene was compared to syntenic regions with significant gain or loss in the canine and equine genomes in order to identify amplified or deleted genes common to human and canine

and/or equine melanoma. This procedure was repeated to compare canine and equine SCNAs.

**Comparison of primary tumors with metastases and recurrences**. Somatic mutations were called in each primary, metastatic, and recurrent tumors, as described above, using a matched normal as the reference sample in each case. The somatic mutations in the primary tumors were then compared to those in the metastatic and recurrent tumors to determine the number of shared and private mutations.

## Data availability

The raw sequencing data are available for download from the European Genome-phenome Archive under study accession EGAS00001001115 (human), and from the European Nucleotide Archive under study accessions ERP013521 (canine) and ERP012934 (equine).

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

## Acknowledgements

This work was funded by grants to D.J.A. from the Wellcome Trust, Cancer Research UK, and the European Research Council under the European Union's Seventh Framework Programme (FP7/2007–2013)/ERC synergy grant agreement n° 319661 COMBATCANCER.

## Author contributions

K.W., L.v.d.W., C.R.S., M.J.A., T.B., P.W.H., G.A.W., and D.J.A. designed the study and wrote the paper. A.F., F.C.-C., S.S., J.M.D., E.P.M., H.W., I.Y., D.R.F., N.J., B.C.B., R.M.P., M.J.A., T.B., and P.W.H. provided DNA samples and clinical data. K.W., I.M., C.D.R.-E., V.I. and M.L.K. performed computational analyses.

## Additional information

**Competing interests:** The authors declare no competing interests.

