## [Peer Review File · Nature Communications]

Reviewers' comments:

Reviewer #1 (Remarks to the Author):

The manuscript submitted by Wong et al reports on their work performing a comparative genomic analysis of 46 human mucosal melanoma cases, 65 canine oral melanoma, and 62 equine melanoma cases. This work represents the largest series of human (as well as canine and equine) mucosal melanoma samples to undergo whole exome sequencing to be reported thus far, with prior published human mucosal melanoma series including under 20 cases each. The authors report some recurrent driver mutations to be shared between species, including NRAS, FAT4, PTPRJ, BRCA2 and PTEN, as well as relevant germline predisposing mutations in humans including BRCA1 and BRCA2. As previously reported, mucosal melanomas were found to have fewer somatic mutations when compared with what is seen in cutaneous disease, with frequent copy number changes in human and canine cases. The uncommon finding of a UV mutational signature in several cases of disease arising in the lip and sinonasal region is of interest. Importantly, through sequencing of matched primary/recurrent and primary/metastatic lesions, findings consistent with pervasive intra-tumour heterogeneity were identified. Overall, the manuscript is well written and the methods well described. My specific comments are as follows:

1. Introduction: Readers may benefit from some description of the classification of equine melanomas within the main text, with some clarification of the definition of "mucosal-like" and "mucocutaneous" melanomas (which is already included in the methods).
2. Page 6, Lines 134: Consider specifying that the 5 "recurrences" were all local or locoregional recurrences.
3. Supplemental Table 1a: It would be useful to include additional clinical information regarding the recurrent and metastatic lesions which were sequenced. Specifically, what was the interval of time between the diagnosis of the primary lesions and the development of the recurrence/distant metastasis?
4. Supplemental Table 1a: Furthermore, as vulvar melanoma arising from hair-bearing regions have clinical and molecular characteristics similar to cutaneous melanoma, it would be helpful to provide further details regarding specific site of origin within the vulva if available.
5. Supplemental Table 1a: The tissue source for some of the metastatic specimens seem to be the same as the primary tumor (eg. PD25645c and PD25645a). Please confirm that all metastasis are in fact distant metastatic specimens and not locally recurrent tumors.
6. Supplemental Table 1a: Patient demographics (gender, race/ethnicity) would also strengthen the table if available.
7. Page 6, Line 159: I would consider explicitly stating that the 3 sinonasal tumor specimens with a UV mutational signature arose from the nasal mucosal (and not from the sinuses).

Reviewer #2 (Remarks to the Author):

Wong et. al. took a cross-species approach to study the genomic alterations of mucosal melanomas. The motivation is interesting and novel in itself. However, the study design seems highly biased and the data analyses are not comprehensive enough, making it difficult to draw strong conclusion.

Major comments:

1. Mucosal melanomas in the three species examined (human, dog, horse) are from quite diverse and different tissue sites. For example, the 65 canine melanomas are all from oral, the 46 human samples are mainly from sinonasal and vulvar sites, and the 28 equine samples are mainly preputial and perineal. Melanomas from different sites actually displayed clear heterogeneity, as depicted from the diverse mutation burdens in equine melanomas. Such a heterogeneous and unbalanced experimental design is highly undesired, especially when the sample sizes (ranging from 28 to 65) are not big enough. Somatic mutations are well known to have a long tail distribution, which are unlikely to be sensitively unraveled with a limited sample size and unbalanced experimental design. Furthermore, any subsequent comparisons and in-depth analyses would be hindered by these design limitations.
2. The copy number estimates from exome-sequencing data could provide useful information from a different aspect. However, the authors seem simply ignored inter-species chromosome alignment before they can compare across the three species (Figure 3). It is recommended to first identify the syntenic regions across the three species and then project the copy number estimates to draw the copy number alteration landscape.

Minor comments:

1. Figure 2 does not provide significantly more information than Figure 1, and should be provided as supplementary data.
2. The frequently mutated genes shown in Figure 1 are better accompanied with mutation significances, as the simple frequency index is known to be biased to longer genes.
3. The status of metastasis and recurrence is better to be annotated in Supplementary Tables 2-3.

Reviewer #3 (Remarks to the Author):

Kim Wong et al describe a comparative cancer genomics study of mucosal melanoma in humans, dogs and canines. The authors have sequenced 46 human primary mucosal melanomas, 65 oral canine primary melanomas, and 62 equine primary melanomas from a variety of sites (mucosal-like or mucocutaneous, cutaneous, or other). These samples also include some recurrences and metastases. This represents an interesting data set to discover the similarities and differences between the mutational landscapes of similar cancer types in 3 different species.

The authors present the status of previously established melanoma driver genes and recurrently mutated genes identified in these data across the three species. With the exception of a few genes unannotated in some species, demonstration or discussion of one-one orthology of genes being considered is notably absent from the manuscript. Clear orthology may seem obvious in some cases, but not all. One-to-one orthology should be checked for each of these genes. Statements or Supplementary Table should be added supporting this.

They also undertake mutational signature analyses on a subset of the human (12) and horse (5) samples using NMF and cosine similarity to compare to the previously identified signatures, but only mention results for the UV signature. Results for this analysis should be presented more completely (as supplementary figures would be fine).

It's unfortunate that of the ~180 samples, only 17 could be used for mutational signature analysis. For the human samples only, would regression of the COSMIC signatures allow you to get below the 100 mutation cutoff applied? Would combining samples within species, as some authors have done, help to isolate other dominant signatures?

The authors compare the copy number gain and losses of each cancer type. This was done by means of frequency plots against chromosomal position in each species. Unfortunately, no attempt was made to integrate these, leading to fairly superficial comparisons and the possibility of missed biological insights. A difficult problem in cancer genomics is identifying the genes whose

amplification or deletion are under selection in regions of large-scale (whole chromosome or whole arm) copy number change. Optimistically, comparative cancer genomics could help here, as genes or groups of genes can be shuffled around the genome between species. For example, based on large-scale alignment and chaining:

- * Human chr8 is orthologous to blocks in dog chromosomes 16, 25, 29 and 13 (which also shows frequent amplification).

- * Equine chr25, which shows moderate levels of recurrent amplification, is orthologous with human chr9 and canine chr11, which generally have different patterns of copy number change. Alternatively, this might be done more easily using gene chromosomal ordering and orthology. To me this is an interesting and potentially powerful idea, and a creative use fo the data that would add a lot to this paper.

The observation that dog oral melanomas may not represent a faithful genetic model for all subtypes of human mucosal melanoma is an important cautionary tale.

The manuscript is generally very well written. The analyses are appropriate, but version numbers are not provided for bioinformatics methods. The analysis code should also be provided to improve the reproducibility of the results.

The manuscript provides potentially useful landscapes of cutaneous melanoma in three species, but I feel it misses some opportunities to make more creative and interesting use of this data. Figure 2 integrates the SNVs, but there is no attempt to integrate copy numbers of genes or orthologous (syntenic) regions. With the exception of Figures 1 and 2, The addition of some further creative analyses as recommended above would enhance the novelty, level of interest and importance of the contribution.

Reviewers' comments:

Reviewer #1 (Remarks to the Author):

The manuscript submitted by Wong et al reports on their work performing a comparative genomic analysis of 46 human mucosal melanoma cases, 65 canine oral melanoma, and 62 equine melanoma cases. This work represents the largest series of human (as well as canine and equine) mucosal melanoma samples to undergo whole exome sequencing to be reported thus far, with prior published human mucosal melanoma series including under 20 cases each. The authors report some recurrent driver mutations to be shared between species, including NRAS, FAT4, PTPRJ, BRCA2 and PTEN, as well as relevant germline predisposing mutations in humans including BRCA1 and BRCA2. As previously reported, mucosal melanomas were found to have fewer somatic mutations when compared with what is seen in cutaneous disease, with frequent copy number changes in human and canine cases. The uncommon finding of a UV mutational signature in several cases of disease arising in the lip and sinonasal region is of interest.

Importantly, through sequencing of matched primary/recurrent and primary/metastatic lesions, findings consistent with pervasive intra-tumour heterogeneity were identified. Overall, the manuscript is well written and the methods well described.

We thank the reviewer for their comments, particularly their appreciation that “*This work represents the largest series of human (as well as canine and equine) mucosal melanoma samples to undergo whole exome sequencing to be reported thus far, with prior published human mucosal melanoma series including under 20 cases each*”. As mucosal melanoma is a very rare malignancy in humans (<1% of melanomas in the US/UK) it has been an enormous effort to collect 46 human cases with matched normal tissue for sequencing. Similarly, collecting well annotated canine and equine cases has been a major undertaking (samples for this study were obtained from 6 different Institutes in 4 different countries).

My specific comments are as follows:

1. Introduction: Readers may benefit from some description of the classification of equine melanomas within the main text, with some clarification of the definition of “mucosal-like” and “mucocutaneous” melanomas (which is already included in the methods).

In our revised manuscript we have clarified these definitions in the first paragraph of the results. We thank the reviewer for this comment.

2. Page 6, Lines 134: Consider specifying that the 5 “recurrences” were all local or locoregional recurrences.

We have clarified this point in the revised manuscript.

3. Supplemental Table 1a: It would be useful to include additional clinical information regarding the recurrent and metastatic lesions which were sequenced. Specifically, what was the interval of time between the diagnosis of the primary lesions and the development of the recurrence/distant metastasis?

Where available, we have added these data in Supplementary Table 1a. As the reviewer will appreciate, the availability of this information varies depending on the hospital. For some cases we don't have access to this information for ethical reasons since revealing the metadata for a case of such a rare cancer may identify the patient.

4. Supplemental Table 1a: Furthermore, as vulvar melanoma arising from hair-bearing regions have clinical and molecular characteristics similar to cutaneous melanoma, it would be helpful to provide further details regarding specific site of origin within the vulva if available.

We have provided this information as requested. Notably, all of the vulvar melanomas in our series had a low mutational load, showed significant copy number gains and losses and also had a profile of driver gene mutations distinct from cutaneous melanoma.

5. Supplemental Table 1a: The tissue source for some of the metastatic specimens seem to be the same as the primary tumor (eg. PD25645c and PD25645a). Please confirm that all metastasis are in fact distant metastatic specimens and not locally recurrent tumors.

We re-examined the clinical records and spoke to the pathologists who provided these cases who then went and re-reviewed these cases again and confirmed that these specimens are metastatic and not locally recurrent tumors. We have corrected the tissue sources in Supplementary Table 1a. All are locoregional mets, with the exception of one distant metastatic tumour forming outside the lymphatic drainage region (lung). We thank the reviewer for their very careful analysis of the clinical data.

6. Supplemental Table 1a: Patient demographics (gender, race/ethnicity) would also strengthen the table if available.

Gender information is present in Supplementary Table 1, under the column header “Sex”. As described in the Results and Methods, to assess ethnicity, we have performed a principal component analysis using variants from the 1000 Genomes Project Phase I populations (Supplementary Figure 1). This analysis revealed that all but 3 of the cases in this study were of European descent, and we have added this information to Supplementary Table 1a.

7. Page 6, Line 159: I would consider explicitly stating that the 3 sinonasal tumor specimens with a UV mutational signature arose from the nasal mucosal (and not from the sinuses).

This is a very good point. We have amended the text accordingly.

We thank Reviewer 1 for their insightful comments.

Reviewer #2 (Remarks to the Author):

Wong et. al. took a cross-species approach to study the genomic alterations of mucosal melanomas. The motivation is interesting and novel in itself. However, the study design seems highly biased and the data analyses are not comprehensive enough, making it difficult to draw strong conclusion.

We thank reviewer 2 for their comments. By de-sparsifying our somatic mutation data and performing a principal component analysis (described below), we show that there is no

association between tissue site and mutational patterns and the suggestion that our study design is “highly biased” is without foundation. Further, the reviewer should be aware that mucosal melanoma is a vanishingly rare malignancy and that 46 human cases is the largest collection ever assembled (the next largest collection is just 19 cases [PMID: 29230811]). Notably, no one has ever studied the exome-wide mutation profile of either canine or equine mucosal melanomas so we, or indeed others, did not know how similar or different these cases would be prior to our analysis. We did not know if human sinonasal cases or cases from the lip would appear genetically more similar to canine oral melanomas when compared to mucosal melanomas from other sites, or if equine cases from the preputial or perineal region would be more like human cases from the urogenital region – as it turns out neither appears to be the cases. We note the reviewer’s comment that our study is novel and have specifically addressed the strengths and potential weaknesses of our approach in the revised manuscript.

Major comments:

1. Mucosal melanomas in the three species examined (human, dog, horse) are from quite diverse and different tissue sites. For example, the 65 canine melanomas are all from oral, the 46 human samples are mainly from sinonasal and vulvar sites, and the 28 equine samples are mainly preputial and perineal. Melonomas from different sites actually displayed clear heterogeneity, as depicted from the diverse mutation burdens in equine melanomas. Such a heterogeneous and unbalanced experimental design is highly undesired, especially when the sample sizes (ranging from 28 to 65) are not big enough.

We thank the reviewer for these comments and have made clear exactly at which sites mucosal melanomas develop in each species, since there seems to be some confusion about the ascertainment of the cases we sequenced. Importantly, our cross-species analysis did not include acral or cutaneous subtypes of melanoma, where clear differences in mutation profiles/driver genes have been previously noted – we focused exclusively on mucosal melanoma. We did sequence horse cutaneous cases, but as we clearly stated in the main text of the manuscript, we used these as a comparator for the horse mucosal cases – not as part of the cross-species analysis. We respond to each of reviewer 2’s comments in detail below.

For example, the 65 canine melanomas are all from oral, the 46 human samples are mainly from sinonasal and vulvar sites, and the 28 equine samples are mainly preputial and perineal.

These are the major sites where humans, dogs and horses develop mucosal melanoma. In humans, while melanomas can develop within any mucosal surface, the vast majority arise in the mucosae of the head and neck (31-55%), vulvovaginal (18-40%) and anorectal (17-24%) regions [PMID: 9781962, 15651058]. Canine melanomas rarely arise at sun-exposed sites, with most occurring in the oral cavity [PMID: 24128326], and there are numerous reports in the literature suggesting that naturally occurring oral melanoma in dogs as a model for human mucosal melanoma [PMID: 24128326, 24112648, 17591290, 23745794]. In horses, the underside of the tail, the perineal and peri-anal regions, and the penis and sheath in males are the most common locations for melanomas to develop [Karen Briggs, “Sarcoids and Melanomas”, *The Horse*, 1999; <http://edelsonequine.com/latest-news/3691105/Equine-Sarcoids-and-Melanomas/2166763>].

To specifically address the comment about tissue site and “bias” we performed the analysis below using the algorithm SAMBAR (Kuijjer *et al.*, 2018 [PMID:29765148]). This analysis provides no support for any relationship between tissue site and mutation profile for either human or equine mucosal melanomas (this analysis is detailed in the Methods of the revised manuscript). Since all the canine cases were oral we did not perform this analysis.

Supplementary Figure 2: Principal component analysis (PCA) of mutation data from human and equine mucosal melanoma samples sourced from various tissue sites. (a) PCA plot of human mutation data de-parsified into biological data pathways. (b) Heatmap of the correlation coefficients (R^2), as indicated by the color key, from PC regression against phenotypic variables in the human and horse dataset. Numbers inside of the heatmap cells are the adjusted P -values of the regression. The first five PCs are listed on the x-axis, and the percentage of variance that is explained by each PC is given inside parentheses. Phenotypic data are derived from Supplementary Table 1. Samples from specific tissues were grouped into tumour sites, which are shown in Figures 1 and 2. Tissues and tumour sites are listed for each sample in Supplementary Table 1. There was no significant correlation between the first 5 PCs and any of these variables.

Melanomas from different sites actually displayed clear heterogeneity, as depicted from the diverse mutation burdens in equine melanomas.

We are aware, and have published extensively, on the fact that cutaneous melanomas (from sun-exposed hair-bearing skin) are different from mucosal melanomas, which is why all of the cross-species analyzes we performed do not include the equine cutaneous cases, and our figures clearly segregate the equine mucosal-like and mucocutaneous samples from tumours from other sites. To reiterate, different types of melanoma (cutaneous vs. mucosal) show heterogeneity in their mutation burden but being mindful of this, our paper focused on mucosal melanomas. Further,

we note some apparent differences in the genes mutated at different tissue sites in humans, but the SAMBAR analysis described above does not support the idea that there are significant differences in the pathways that are altered in tumours from different tissue sites.

Such a heterogeneous and unbalanced experimental design is highly undesired, especially when the sample sizes (ranging from 28 to 65) are not big enough.

As above, and as noted by reviewer 1, this is the largest study of its kind and more than twice the size of any other human series. Our study is also the first to analyse the exome-wide mutation profiles of equine and canine cases and, as such, we are somewhat perplexed by this comment – which is made without any statistical support, references or reasoning. As stated throughout the manuscript our aim was to compare mucosal cases from the different species to determine how different or similar they are. We have commented further on the sample numbers below.

Somatic mutations are well known to have a long tail distribution, which are unlikely to be sensitively unraveled with a limited sample size and unbalanced experimental design.

Furthermore, any subsequent comparisons and in-depth analyses would be hindered by these design limitations.

Our study is bigger than virtually all rare cancer sequencing studies published in Nature Communications this year. For example, pineoblastoma (PMID: 30030436), hepato-cholangiocarcinoma (PMID:29497050) and adenomyoepitheliomas (PMID:29739933) had 23, 15 and 43 samples, respectively. Further, to exhaustively characterise all possible drivers in a malignancy, estimates suggest that >2000 tumours would be required (PMID: 24390350). This is simply not feasible in a cancer with a population frequency of less than 1/100,000 people (and is yet to be performed for any cancer). That said, we have collected the largest series of mucosal melanomas from across the globe. We challenge the view that any other experimental design would be possible or feasible. Further, we show that across and within species the mutations we identify coalesce into key pathways, and we also identify recurrently mutated genes. This clearly illustrates that our approach and study design is informative.

2. The copy number estimates from exome-sequencing data could provide useful information from a different aspect. However, the authors seem simply ignored inter-species chromosome

alignment before they can compare across the three species (Figure 3). It is recommended to first identify the syntenic regions across the three species and then project the copy number estimates to draw the copy number alteration landscape.

In our revised manuscript we performed additional analysis to examine the cross-species similarities and differences in copy number landscapes in mucosal melanomas. We obtained syntenic regions between species from the Ensembl Compara database and identified syntenic regions with recurrent copy number gains or losses across species. We also used GISTIC 2.0 and STAC to identify focal and large-scale alterations that occur at a statistically significant rate above background. Using these approaches, we identified known and potentially novel driver genes within these regions and discuss similarities and differences between species. These data are presented in Table 1, Figure 5 and Supplementary Tables 5-7 of the revised manuscript.

Of note, the original Figure 3a was not meant to be a cross-species comparison of copy number alterations. We have corrected the wording of the first sentence in this section to clarify this. Figure 3a shows the frequencies of specific copy number changes within each species (i.e. it provides an indication of the overall somatic copy number landscape within each species), and, importantly, we stated in the text that we were able to replicate the findings of specific recurrent copy number alterations from array-based studies in human (PMID: 16291983) and dog (PMID: 25511566) and report a novel finding recurrent SCNAs in in equine mucosal melanomas.

Minor comments:

1. Figure 2 does not provide significantly more information than Figure 1, and should be provided as supplementary data.

Figure 1 shows mutations in a selected set of established melanoma driver genes based on previous studies (a supervised comparison). In Figure 2, we leveraged the power of cross-species comparison to search for novel genes of interest by identifying recurrently mutated genes in at least 2 of the 3 species. The analysis revealed changes in genes such as *TCF7L1* and *FAT4*, which have not been reported previously. This figure also helps to demonstrate that while mutations in established drivers such as *NRAS* and *TP53* are common to mucosal melanomas from all 3 species, there are several cancer-associated genes altered in only canine and human cases.

2. The frequently mutated genes shown in Figure 1 are better accompanied with mutation significances, as the simple frequency index is known to be biased to longer genes.

All of the genes shown in Figure 1 are established melanoma genes. The point of this figure is to illustrate which of these genes are mutated in melanomas from each species and to give a sense of the mutation frequency. Specifically, the text indicates that many of the mutations in these genes are at hotspot positions. Thus, we have used a standard approach as deployed by many other investigators (for example PMID: 2846782/28467829 and 30178487) including TCGA (PMID: 26091043).

3. The status of metastasis and recurrence is better to be annotated in Supplementary Tables 2-3.

Supplementary Tables 2 and 3 (now Supplementary Tables 2a and 2b in the revised manuscript) provide a list the somatic mutations in each human and canine sample. We interpret the reviewer's comment to mean that they are asking us to indicate which mutations are from a primary, metastasis or recurrence. We have added this information as a column in these tables as requested.

We thank Reviewer 2 for their comments on our manuscript.

Reviewer #3 (Remarks to the Author):

Kim Wong et al describe a comparative cancer genomics study of mucosal melanoma in humans, dogs and canines. The authors have sequenced 46 human primary mucosal melanomas, 65 oral canine primary melanomas, and 62 equine primary melanomas from a variety of sites (mucosal-like or mucocutaneous, cutaneous, or other). These samples also include some recurrences and metastases. This represents an interesting data set to discover the similarities and differences between the mutational landscapes of similar cancer types in 3 different species.

We thank Reviewer 3 for their positive comments on our study.

The authors present the status of previously established melanoma driver genes and recurrently mutated genes identified in these data across the three species. With the exception of a few genes unannotated in some species, demonstration or discussion of one-one orthology of genes being considered is notably absent from the manuscript. Clear orthology may seem obvious in some cases, but not all. One-to-one orthology should be checked for each of these genes. Statements or Supplementary Table should be added supporting this.

We have included a new Supplementary Table 4, which includes all orthology relationships for genes in Figures 1, 2, and 5 from our revised manuscript, the sequence identities between orthologs, GOC scores and orthology confidence scores from Ensembl release v91. For almost all of the annotated genes we present in Figures 1, 2 and 5, there is 1-to-1 orthology between human and canine genes, and human and equine genes (as defined by Ensembl Compara). The exceptions are canine *MAP2K2*, canine and equine *DNAH5*, and canine *CDKN2B*.

They also undertake mutational signature analyses on a subset of the human (12) and horse (5) samples using NMF and cosine similarity to compare to the previously identified signatures, but only mention results for the UV signature. Results for this analysis should be presented more completely (as supplementary figures would be fine).

In the signature analysis we applied, there are 96 possible mutation contexts, therefore when many samples with few mutations are used, the motif matrix is sparse, and the per-sample results are unreliable. Thus, we focused our analysis on the samples with >100 mutations to identify specific signatures in each sample. Only two signatures were identified, the UV signature (COSMIC signature 7) and COSMIC signature 1. We have amended to text to specify this. Since signature 1 is seen in all cancer types and is related to age and 5mC deamination, we did not focus on this finding in our original results or discussion.

In an attempt to identify more signatures, as suggested by the reviewer, we grouped samples within species and pooled mutations to perform a signature analysis (discussed further below). No further signatures were discovered. We have discussed this analysis in the revised manuscript and provide details in the Methods and Supplementary Table 3..

It's unfortunate that of the ~180 samples, only 17 could be used for mutational signature analysis. For the human samples only, would regression of the COSMIC signatures allow you to get below the 100 mutation cutoff applied? Would combining samples within species, as some authors have done, help to isolate other dominant signatures?

As suggested by the reviewer, we have used a pooling strategy for tumours within a species in an attempt to identify additional signatures. Because a minimum of 2 groups is required for analysis by the R package SomaticSignatures, we grouped the human samples by tumour site (sinonasal and oral tumours in one group, anogenital and vulvar in the other), and the equine samples by tumour type (mucosal-like/mucocutaneous in one group, cutaneous and "other" in the second group). For the human analysis, recurrences and metastases were included to increase sample and mutation numbers; none were available for the equine data set. No signatures in addition to signatures 1 and 7 were identified, although one signature had weak cosine similarity so signature 5, a signature that is seen in all cancer types. Since the canine samples were all oral melanomas, we arbitrarily grouped the primary samples and 6 matched metastases into 2 groups (35 and 36 samples) to perform a signature analysis. Only signature 1 was identified. We provide in our revised manuscript a new Supplementary Table 3 which shows the results of our mutation signature analysis. (We note that there are now 16 samples with >100 mutations as we previously used a slightly less filtered mutation call set to perform signature analysis in order to maximize the input to SomaticSignatures.)

The authors compare the copy number gain and losses of each cancer type. This was done by means of frequency plots against chromosomal position in each species. Unfortunately, no attempt was made to integrate these, leading to fairly superficial comparisons and the possibility of missed biological insights.

In our revised manuscript we have performed the analysis suggested below. Of note Figure 3 was not meant to be a cross-species comparison of copy number changes. Figure 3 shows the frequencies of specific copy number changes in each species.

A difficult problem in cancer genomics is identifying the genes whose amplification or deletion are under selection in regions of large-scale (whole chromosome or whole arm) copy number

change. Optimistically, comparative cancer genomics could help here, as genes or groups of genes can be shuffled around the genome between species. For example, based on large-scale alignment and chaining:

* Human chr8 is orthologous to blocks in dog chromosomes 16, 25, 29 and 13 (which also shows frequent amplification).

* Equine chr25, which shows moderate levels of recurrent amplification, is orthologous with human chr9 and canine chr11, which generally have different patterns of copy number change. Alternatively, this might be done more easily using gene chromosomal ordering and orthology. To me this is an interesting and potentially powerful idea, and a creative use of the data that would add a lot to this paper.

We completely agree and it was remiss of us not to include this analysis. The same comment was made by reviewer #1 (Q2) and we provide the response again here. We obtained syntenic regions between species from the Ensembl Compara database and identified syntenic regions with recurrent copy number gains or losses across species. We also used GISTIC 2.0 and STAC to identify focal and large-scale alterations that occur at a statistically significant rate above the background. Using these approaches, we identified known and potentially novel driver genes within these regions and discuss similarities and differences between species. These data are presented in Table 1, Figure 5 and Supplementary Tables 5-7 of the revised manuscript.

The observation that dog oral melanomas may not represent a faithful genetic model for all subtypes of human mucosal melanoma is an important cautionary tale.

The manuscript is generally very well written.

We thank the reviewer for their positive comments on our work.

The analyses are appropriate, but version numbers are not provided for bioinformatics methods.

The analysis code should also be provided to improve the reproducibility of the results.

We have provided the version numbers in the revised manuscript as suggested.

The manuscript provides potentially useful landscapes of cutaneous melanoma in three species, but I feel it misses some opportunities to make more creative and interesting use of this data. Figure 2 integrates the SNVs, but there is no attempt to integrate copy numbers of genes or orthologous (syntenic) regions. With the exception of Figures 1 and 2, The addition of some

further creative analyses as recommended above would enhance the novelty, level of interest and importance of the contribution.

As detailed above, we have performed this analysis and provide additional text in the Result and Discussion sections, and include a new Table 1, Figure 5 and Supplementary Tables 5-7 in the revised manuscript. We agree that it enhances the analysis and thank the reviewer for their helpful comment.

Reviewers' comments:

Reviewer #2 (Remarks to the Author):

In the revised manuscript of Wong et. al., significant efforts have been made to refine the study. They have addressed the potential tissue site heterogeneity when interpreting the mutational landscape, and also added the syntenic analysis for inter-species comparison of the global CNAs (Figure 5, etc.).

1. Although the sample sizes for each tissue site are quite small (e.g. 17 sinonasal, 8 vaginal, 12 vulvar, 4 anal, 1 urethral, 3 oral in human melanomas; 65 oral in dog melanomas; 3 perianal, 7 perineal, 8 preputial, 5 tail in horse melanomas), pooled analysis indeed can provide useful insights of the general picture of mucosal melanomas, as long as this "pooling" indeed makes sense. To comment on the PCA analysis of pathway-level mutation data, it won't be surprising to find no strong associations to the many tissue sites with few samples. My real concern is, from the data present in text and Figure 1, roughly three "clusters" can be seen: a major cluster with frequent NRAS or TP53 muts (human vaginal/vulvar/anal/urethral + dog oral + horse tail, ~95 samples), a second cluster with frequent NRAS but not TP53 mut (human sinonasal, 17 samples), a third cluster with rare NRAS or TP53 muts (horse perianal/perineal/preputial, 18 samples). So it seems not yet the time to conclude that "melanomas arising from different mucosal sites do not represent different subtypes" (line 216, p.8). Future studies might examine whether these potential subtypes have different metastatic capabilities and prognosis. In addition, any specific germline muts or CNAs in the human sinonasal samples to silence the TP53-related pathways?

2. Table 2: does not include patient ID (some mentioned in the text), which might be useful to know. Alternatively, the authors may wish to provide a new supplementary table of detailed germline muts in each patient.

Reviewer #3 (Remarks to the Author):

The authors have done a good job of addressing my own and the other reviewers comments.

The comparative use of data between species has been improved. I think that Supp Figure 3 is fascinating, and is an excellent contribution to the manuscript.

I'd also like to acknowledge and highlight the authors point about mucosal melanoma being rare and in light of this, the size of the cohort is actually substantial and the design appropriate.

A minor point is that two different spellings of orthologs/orthologues have been used. This should be corrected.

The manuscript makes an important contribution and should now be accepted.

Reviewer #2 (Remarks to the Author):

In the revised manuscript of Wong et. al., significant efforts have been made to refine the study. They have addressed the potential tissue site heterogeneity when interpreting the mutational landscape, and also added the syntenic analysis for inter-species comparison of the global CNAs (Figure 5, etc.).

We thank reviewer 2 for acknowledging the significant effort we have invested in revising the manuscript and performing the cross-species comparison of SCNAs as suggested.

1. Although the sample sizes for each tissue site are quite small (e.g. 17 sinonasal, 8 vaginal, 12 vulvar, 4 anal, 1 urethral, 3 oral in human melanomas; 65 oral in dog melanomas; 3 perianal, 7 perineal, 8 preputial, 5 tail in horse melanomas), pooled analysis indeed can provide useful insights of the general picture of mucosal melanomas, as long as this "pooling" indeed makes sense. To comment on the PCA analysis of pathway-level mutation data, it won't be surprising to find no strong associations to the many tissue sites with few samples. My real concern is, from the data present in text and Figure 1, roughly three "clusters" can be seen: a major cluster with frequent NRAS or TP53 muts (human vaginal/vulvar/anal/urethral + dog oral + horse tail, ~95 samples), a second cluster with frequent NRAS but not TP53 mut (human sinonasal, 17 samples), a third cluster with rare NRAS or TP53 muts (horse perianal/perineal/preputial, 18 samples). So it seems not yet the time to conclude that "melanomas arising from different mucosal sites do not represent different subtypes" (line 216, p.8). Future studies might examine whether these potential subtypes have different metastatic capabilities and prognosis. In addition, any specific germline muts or CNAs in the human sinonasal samples to silence the TP53-related pathways?

We appreciate this thoughtful comment and have amended the manuscript to make clear that with a larger sample size and data aggregation it may be possible to identify subtypes not evident from our sample collection. To address the question of other germline mutations or CNAs that could silence the *TP53* pathway in sinonasal tumours in the absence of somatic *TP53* mutations, we searched for variants in 30 genes previously implicated in *TP53* regulation [Kruse and Gu *et al.*, Cell (PMID: 19450511)]. No known pathogenic variants were identified, nor were any of these genes recurrently altered with rare or novel germline variants. The germline CNA calls we generated (using CNVkit (PMID: 27100738) and Control-Freec (PMID: 22155870) were unreliable due to excessive noise in the data. It is possible that other mechanisms, such as those that act post-transcriptionally or translationally to silence *TP53* function, may be operative and this could be investigated in future studies.

2. Table 2: does not include patient ID (some mentioned in the text), which might be useful to know. Alternatively, the authors may wish to provide a new supplementary table of detailed germline muts in each patient.

We have amended Table 2 as suggested to include sample IDs so that readers can link germline mutations to other patient data.

We thank Reviewer 2 for their assistance in improving our manuscript.

Reviewer #3 (Remarks to the Author):

The authors have done a good job of addressing my own and the other reviewers comments.

The comparative use of data between species has been improved. I think that Supp Figure 3 is fascinating, and is an excellent contribution to the manuscript.

We thank reviewer 3 for these positive comments and their assistance in revising our manuscript. We feel that their contributions have greatly improved the paper.

I'd also like to acknowledge and highlight the authors point about mucosal melanoma being rare and in light of this, the size of the cohort is actually substantial and the design appropriate.

Thank you. It has taken a lot of work to collect these samples for sequencing.

A minor point is that two different spellings of orthologs/orthologues have been used. This should be corrected.

We have corrected this error. House style for *Nature Communications* seems to be "orthologs" so we have used this spelling.

The manuscript makes an important contribution and should now be accepted.

We thank Reviewer 3 for reviewing our manuscript and for their constructive comments.

Reviewer #2 (Remarks to the Author):

The authors have answered my questions and I feel it's acceptable for the publication now.

REVIEWERS' COMMENTS: Reviewer #2 (Remarks to the Author): The authors have answered my questions and I feel it's acceptable for the publication now. We thank reviewer 2 for their assistance in improving our manuscript.